# APPROXIMATING MULTIPLE ROBUST OPTIMIZATION SOLUTIONS IN ONE PASS VIA PROXIMAL POINT METHODS

## ABSTRACT

Robust optimization provides a principled and unified framework to model many problems in modern operations research and computer science applications, such as risk measures minimization and adversarially robust machine learning. To use a robust solution (e.g., to implement an investment portfolio or perform robust machine learning inference), the user has to *a priori* decide the trade-off between efficiency (nominal performance) and robustness (worst-case performance) of the solution by choosing the uncertainty level hyperparameters. In many applications, this amounts to solving the problem many times and comparing them, each from a different hyperparameter setting. This makes robust optimization practically cumbersome or even intractable. We present a novel procedure based on the proximal point method (PPM) to approximate many Pareto-efficient robust solutions using the PPM trajectory. Compared with the existing method with computation cost $N \times T_{\mathrm{RC}}$, the cost of our method is $T_{\mathrm{RC}} + (N - 1) \times T_{\widehat{\mathrm{PPM}}}$, where $N$ is the number of robust solutions to be generated, $T_{\mathrm{RC}}$ is the cost of solving a single robust optimization problem, and $T_{\widehat{\mathrm{PPM}}}$ is cost of a single step of an approximate PPM. We prove exact PPM can produce exact Pareto efficient robust solutions for a class of robust linear optimization problems. For robust optimization problems with nonlinear and differentiable objective functions, compared with the existing method, our method equipped with first-order approximate PPMs is computationally cheaper and generates robust solutions with comparable performance.

## 1 INTRODUCTION

One of the main obstacles in deploying robust optimization models for real-world decision-making under uncertainty is determining an appropriate trade-off between nominal performance and worst-case performance before the uncertainty is realized. For example, in portfolio optimization, setting the level of risk aversion is challenging. Similarly, in adversarial machine learning, choosing the adversarial perturbation set parameters to balance average accuracy versus accuracy under adversarial attacks is a critical decision. Decision makers often need to obtain and test multiple robust solutions, each corresponding to a different solution on the efficiency-robustness Pareto frontier, before deciding which one to deploy.

To obtain multiple robust solutions, decision makers adjust hyperparameters such as the shape and size (radius) of the uncertainty set (Soyster, 1973; Ben-Tal & Nemirovski, 1999; Bertsimas & Sim, 2004) or by employing globalized robust optimization approaches that use penalty functions and coefficient hyperparameters (Ben-Tal et al., 2006; 2009). Consequently, generating each robust solution demands solving a different instance of the robust optimization problem, which is generally more computationally expensive than solving their deterministic counterparts. For instance, adversarial training is more expensive than traditional training because it involves solving an inner maximization problem to find adversarial attacks. In theory, the robust counterpart of linear programs (LPs) with ellipsoidal uncertainty sets becomes second-order cone programs (SOCPs), and the robust counterpart of uncertain SOCPs becomes semidefinite programs (SDPs). As a result, finding the right robust solution can be prohibitively costly in practice, let alone finding many of them to compare.

**Contributions.** To address this challenge, we propose a new way of thinking about this problem, along with a novel proximal point method (PPM)- based algorithm to approximate efficiency-robustness Pareto efficient robust solutions. Compared with the cost of the existing approach of $N \times T_{\text{RC}}$, the computational cost of our method is $T_{\text{RC}} + (N-1) \times T_{\widehat{\text{PPM}}}$, where $N$ is the number of robust solutions to be generated, $T_{\text{RC}}$ is the cost of solving a single robust optimization problem, and $T_{\widehat{\text{PPM}}}$ is the cost of a single step of an approximate PPM. Specifically, the procedure first solves for the "most-robust" solution, then uses it as the starting point to perform approximate PPM updates towards the "least-robust" (deterministic counterpart) solution. We discover, intriguingly, that the proximal point trajectory approximates the set of Pareto efficient robust solutions that we want to obtain in the first place.

We prove that for robust LPs with uncertain objective functions under the simplex decision domain and ellipsoidal uncertainty sets, the proximal point trajectory are *exactly* Pareto efficient robust solutions. For robust LPs with a random polyhedron domain, we prove that with high probability, the performances of the Pareto efficient robust solutions are bounded by the performances of two proximal point trajectories. To validate the theoretical results for constrained robust LPs, we conduct numerical experiments on portfolio optimization. To demonstrate the computation efficiency of our method for robust optimization problems with nonlinear and differentiable objective functions, we apply our method to adversarially robust deep learning.

**Related Work.** Given the problem we study, our work is closely related to multiple literatures. Here we review each literature and compare each with our work.

*Proximal Point Method.* In the continuous optimization literature, the interest in PPM (Martinet (1970); Rockafellar (1976b)) has been predominantly in its convergence properties (Rockafellar (1976a); Güler (1991); Ferris (1991); Parikh et al. (2014); Beck (2017)) and its role as a theoretical framework for analyzing the convergence of other optimization algorithms (e.g., gradient descent, Extra-gradient method, optimistic gradient method, and Nesterov's Accelerated gradient method can all be analyzed as approximations of PPM (Ahn & Sra (2022); Mokhtari et al. (2020b)). Our work fundamentally deviates from the literature and focuses on studying the entire trajectory of (approximate) PPM as (approximate) robust optimization solutions.

*Implicit gradient regularization.* Literature has shown iterates of gradient methods when minimizing a loss function alone, provide implicit regularization. Different from the literature (Ali et al. (2020); Wu et al. (2020); Suggala et al. (2018); Barrett & Dherin (2020); Ji et al. (2020); Sun et al. (2023); Li et al. (2021); Ji & Telgarsky (2019); Wang et al. (2023)) that tends to be descriptive and focused on unconstrained problems, as an intermediate result towards our main Theorem, we give a new, direct and constructive proof for the implicit regularization of PPM under the constrained setting. Our work is also closely related to the literature on the equivalence between regularization/risk measure and robustness. It has been established that regularized learning is equivalent to robust optimization (El Ghaoui & Lebret (1997); Xu et al. (2009); Shafieezadeh Abadeh et al. (2015); Mohajerin Esfahani & Kuhn (2018)). Moreover, risk measure minimization is equivalent to robust optimization (Freund (1987); Natarajan et al. (2009b)). Our work establishes a correspondence between the uncertainty set, regularization/risk measure, and PPM distance-generating function.

*Robust Optimization.* Existing work on calibrating the uncertainty set radius typically relies on probabilistic guarantees, namely, prior bounds on solution robustness (Bertsimas et al. (2021); Mohajerin Esfahani & Kuhn (2018); Blanchet et al. (2019a)). However, such approaches assume prior knowledge of the uncertainty distribution or observations on the uncertainty, which can be unavailable in practice. Even with distributional information or data on the uncertainty, the resulting radius typically leads to overly conservative robust decisions. An alternative is to generate multiple robust solutions under varying radii before selecting the solution for deployment. To the best of our knowledge, no methods exist for generating multiple robust solutions other than solving the robust optimization problem multiple times. Our work shows for a class of robust LPs, the exact PPM iterates are exact robust solutions; for robust optimization with nonlinear and differentiable objectives, approximate PPM iterates can serve as computationally cheap approximate robust solutions.

## 2 PROBLEM SETTING AND MAIN IDEA

### 2.1 NOTATIONS

We denote the dual norm of $\|\cdot\|$ as $\|\cdot\|_*$, defined as $\|z\|_* = \sup\{z^\top x : \|x\| \leq 1\}$. The $p$-norm, $\|\cdot\|_p$, is given by $\|x\|_p = \left(\sum_{i=1}^n |x_i|^p\right)^{1/p}$. The infinity norm, $\|\cdot\|_\infty$, is defined as $\|x\|_\infty = \max_{i\in[n]} |x_i|$. The ball of radius $r$ around $x$ with respect to the norm $\|\cdot\|$ is represented by $\mathbb{B}_{\|\cdot\|}(x,r) = \{z \in \mathbb{R}^n : \|z-x\| \leq r\}$, and the ball of radius $r$ around the origin is denoted as $\mathbb{B}_{\|\cdot\|}(r) = \{z \in \mathbb{R}^n : \|z\| \leq r\}$. The projection operator onto the set $\mathcal{X}$ is denoted as $\Pi_{\mathcal{X}}(x) = \arg\min_{x'\in\mathcal{X}} \|x' - x\|_2^2$. Finally, $e$ represents a vector of ones.

### 2.2 PROBLEM SETTING

We start with a constrained optimization problem with parameters $a$ in the objective function:

$$\min_{x\in\mathcal{X}} f(x,a).$$

The decision variable $x \in \mathbb{R}^n$ is subject to a compact and convex domain $\mathcal{X}$, $a \in \mathbb{R}^n$ is an uncertain vector, which is only known to belong to an uncertainty set $\mathcal{U}$. The robust counterpart is

$$(\text{RC}) \quad \min_{x\in\mathcal{X}} \max_{a\in\mathcal{U}} f(x,a), \tag{1}$$

where the uncertainty set $\mathcal{U}$ takes the following form:

$$\mathcal{U} = \{a_0 + \xi : \xi \in \Xi \subset \mathbb{R}^n\}.$$

Here $a_0$ is some fixed nominal vector, $\xi$ can be interpreted as the perturbation, and $\Xi$ is a nonempty compact convex set. In the case that $\Xi$ is empty, (RC) reduces to a nominal optimization problem with no uncertainty:

$$(\text{P}) \quad \min_{x\in\mathcal{X}} f(x,a_0).$$

Consequently, we define the efficient decision, $x_E$ as the optimizer to problem (P) and the robust decision, $x_R$ as the optimizer to problem (RC), i.e.,

$$x_E := \arg\min_{x\in\mathcal{X}} f(x,a_0), \qquad x_R := \arg\min_{x\in\mathcal{X}} \max_{a\in\mathcal{U}} f(x,a).$$

For any $x \in \mathcal{X}$, its efficiency, $E(x)$ and robustness, $R(x)$ is defined respectively as its performance under nominal or worst-case uncertainty defined as

$$E(x) := -f(x,a_0), \qquad R(x) := -\max_{a\in\mathcal{U}} f(x,a).$$

It is easy to verify that the following inequalities hold

$$E(x_E) \geq E(x_R), \qquad R(x_R) \geq R(x_E).$$

In practice, the trade-off between the efficiency and robustness of robust solutions can be controlled by adjusting the size of the uncertainty set. Specifically, we define an efficiency-robustness Pareto efficient robust solution with a nonempty compact convex uncertainty set $\mathcal{V}$ such that $a_0 \in \mathcal{V} \subset \mathcal{U}$ as

$$x_{PE}(\mathcal{V}) := \arg\min_{x\in\mathcal{X}} \max_{a\in\mathcal{V}} f(x,a).$$

For instance, under norm-ball uncertainty sets, we define the set of efficiency-robustness Pareto efficient robust solutions as the set of robust solutions generated while adjusting the radius of the norm-ball uncertainty set:

$$\left\{x_{PE}(r) := \arg\min_{x\in\mathcal{X}} \max_{\xi\in\mathbb{B}_{\|\cdot\|}(r)} f(x, a_0 + \xi) : r \in (0,\infty)\right\}.$$

Comparing with existing notions of Pareto efficiency of robust solutions (Iancu & Trichakis (2014)) which requires no other robust solutions that perform as least as well across all uncertainty realizations and better for some uncertainty realizations, we relax the requirement and trade-off worst-case performance with the performance under a nominal uncertainty realization.

## 2.3 MAIN IDEA

If the uncertainty set radius $r$ for $\Xi$ is set too large, then a solution $x_{\mathrm{PE}}(r)$ of (RC) tends to be overly conservative and gives bad average case performance, i.e., low $\mathrm{E}(x_{\mathrm{PE}}(r))$. If the radius is set too small, then the solution may not perform reliably under large out-of-sample perturbations or adversarial attacks $\xi$, i.e., resulting in low $\mathrm{R}(x_{\mathrm{PE}}(r))$. What is the right $r$?

Existing literature (Mohajerin Esfahani & Kuhn (2018); Blanchet et al. (2019b;a)) points to ways to choose $r$ based on statistical theory, using observed perturbation data $\xi$ to estimate $\Xi$. But in practice, we may not have any data about $\Xi$. Even if we do, such designs are known in the robust optimization community to be overly conservative. The only practical option is to solve the problem (RC) many times, each with a different $r$, and compare the solutions in terms of their average case and worst-case performances.

Our main contribution is proposing a completely new way of thinking about this problem. Instead of solving (RC) $N$ times where $N$ is the number of solutions we want to compare, we can approximate these Pareto efficient robust solutions within two algorithmic passes. The first pass is to obtain a robust solution $x_{\mathrm{PE}}(r_{\max})$ for (RC) with a large value $r = r_{\max}$. Then we use $x_{\mathrm{PE}}(r_{\max})$ as the initial point of an iterative algorithm (approximate PPM), to solve for the problem (RC) with $r = 0$ (i.e., the nominal problem (P)). The intermediate approximate PPM iterates for $x$ provide a reasonable, sometimes perfect, representation of the Pareto efficient robust solutions.

## 3 TECHNICAL PRELIMINARIES

### 3.1 GENERALIZED PROXIMAL POINT METHODS AND THE CENTRAL PATHS

Here we introduce the generalized proximal point method and the central path for the nominal problem (P). Importantly, we introduce a preliminary result toward the proof of our main theorem, which is the equivalence between the sequence generated by the proximal point method for a linear problem (P) and the central path of the linear problem (P).

We begin with the introduction of the generalized proximal point method where a Bregman distance is in place of the usual Euclidean distance. First, we define the Bregman distance. We assume the distance-generating function $\varphi : \mathcal{X} \to \mathbb{R}$ satisfies the following technical assumptions:

A1.  $\varphi$ is strictly convex, closed, and continuously differentiable in $\mathcal{X}$.

A2.  If $\{x_k\}$ is a sequence in $\mathcal{X}$ which converges to a point $\overline{x}$ in the boundary of $\mathcal{X}$, and $y$ is any point in $\mathcal{X}$, then $\lim_{k\to\infty}\langle \nabla h(x_k), y - x_k \rangle = -\infty$.

The Bregman distance $D_\varphi : \mathcal{X} \times \mathcal{X} \to \mathbb{R}$ w.r.t. $\varphi$ is defined as

$$D_\varphi(x, y) = \varphi(x) - \varphi(y) - \langle \nabla\varphi(y), x - y \rangle.$$

**Example 1.**

- *Let $\varphi(x) = \|x\|_2^2$, then $D_\varphi(x, y) = \|x - y\|_2^2$.*

- *Let $\varphi(x) = \langle x, \Sigma x \rangle$, then $D_\varphi(x, y) = \langle x - y, \Sigma(x - y) \rangle$.*

Let $x_0 \in \mathcal{X}$, the generalized proximal point method for solving the problem (P) generates a sequence $\{x_k\} \in \mathcal{X}$ as

$$x_{k+1} = \arg\min_{x\in\mathcal{X}} f(x, a_0) + \lambda_k D_\varphi(x, x_k)$$

where $\{\lambda_k\} \in \mathbb{R}_{++}$ satisfies $\sum_{k=0}^\infty \lambda_k^{-1} = +\infty$.

The central path of the problem (P) with barrier function induced by the Bregman distance as $D_\varphi(\cdot, x_0)$ is defined as $\{x(\omega) : \omega \in (0, \infty)\}$ and

$$x(\omega) = \arg\min_{x\in\mathcal{X}} f(x, a_0) + \omega D_\varphi(x, x_0).$$

If $x_0 = x_{\mathrm{R}}$, the central path can be interpreted as a set of robust solutions that trade-off between efficiency and robustness. Specifically, by adjusting $\omega$, the central path solutions trade off efficiency,

$f(x, a_0)$ with the Bregman divergence to the most robust solution, $D_\varphi(x, x_\mathrm{R})$. The following result states for a linear problem (P), the proximal point method sequence and the central path are equivalent, i.e., a proximal point sequence initialized by $x_0 = x_\mathrm{R}$ is contained in the central path with $x_0 = x_\mathrm{R}$.

**Proposition 1** (**Theorem 3 in Iusem et al. (1999)**). *For problem (P) with linear objective functions $f(x, a_0) := \langle a_0, x \rangle$ and a closed and convex polyhedron domain, $\mathcal{X} := \{x \in \mathbb{R}^n : Ax \le b\}$. Assume that $\varphi$ satisfies A1 and A2. Let $\{x(\omega) : \omega \in (0, \infty)\}$ by the central path of the problem (P) w.r.t. $D_\varphi(\cdot, x_0)$ and let $\{x_k\}$ be the proximal point method sequence for the problem (P) with Bregman distance $D_\varphi$. If the sequence $\omega_k$ is defined as*

$$\omega_k = \left( \sum_{j=0}^{k-1} \lambda_j^{-1} \right)^{-1}, \quad \text{for } k = 1, 2, \ldots,$$

*then*

$$x_k = x(\omega_k), \quad \text{for } k = 1, 2, \ldots.$$

### 3.2 Correspondence between Robust Optimization and Risk Measure Minimization

Our analysis also draws from the deep connection between robust optimization and risk measure minimization. Specifically, it has been shown that risk measures can be mapped explicitly to robust optimization uncertainty sets and vice versa (Natarajan et al. (2009a); Bertsimas & Brown (2009)). In particular, we utilize the following correspondence between the robust optimization ellipsoidal uncertainty set and the mean-standard deviation risk measure.

**Lemma 1** (**Correspondence between Ellipsoidal Robust Linear Optimization and and Mean–Variance Risk Minimization**). *Under some closed convex polyhedron domain $\mathcal{X} \in \mathbb{R}^n$ and ellipsoidal uncertainty set $\Xi(\alpha) = \left\{ \xi \in \mathbb{R}^n : \|\Sigma^{-1/2}\xi\|_2 \le \alpha \right\}$ where $\Sigma \in \mathbb{R}^{n \times n}$ is a symmetric matrix, The robust optimization problem*

$$\min_{x \in \mathcal{X}} \max_{\xi \in \Xi(\alpha)} \langle a_0 + \xi, x \rangle \tag{2}$$

*is equivalent to the following mean-standard deviation risk measure minimization problem*

$$\min_{x \in \mathcal{X}} \langle a_0, x \rangle + \alpha \sqrt{\langle x, \Sigma x \rangle}. \tag{3}$$

The proof of Lemma 1 is presented in Appendix B.

## 4 Theory: Equivalence between the Pareto efficient Robust Solutions and the Proximal Point Method Trajectory

In this section, we show that under the simplex domain and ellipsoidal uncertainty set, a set of Pareto efficient robust solutions to uncertain linear optimization problems with uncertainty in the objective function can be obtained exactly as a proximal point method trajectory. Further, for random problem instances with random polyhedron domains, we show that the performances of the Pareto efficient solutions are bounded probabilistically in between the performances of two proximal point method trajectories. The main implication of our result is the following: instead of solving a different instance of a robust optimization problem to arrive at each solution on the efficiency-robustness Pareto frontier, an entire "menu" of solutions on the efficiency-robustness Pareto frontier can be obtained approximately, under some condition exactly in a single pass via gradient methods.

### 4.1 Exact Result: Simplex Domain and Ellipsoidal Uncertainty Set

We first consider the case of Pareto efficient robust solutions to robust linear optimization problems with uncertain objective functions under the simplex domain and ellipsoidal uncertainty sets. We show a proximal point sequence is contained within the set of Pareto efficient robust solutions. In particular, we have the following theorem.

**Theorem 1** (**Correspondence Between Pareto-Efficiency Robust Solutions and the Proximal Point Trajectory**). *Under linear objective functions, $f(x, a) := \langle a, x \rangle$. Let $\{x_{\mathrm{PE}}(\alpha) : \alpha > 0\}$ be the set of Pareto efficient robust solutions under simplex domain $\Delta^n = \{x \in \mathbb{R}^n_+ : \langle e, x \rangle = 1\}$ and ellipsoidal uncertainty set $\Xi(\alpha) = \{\xi \in \mathbb{R}^n : \|\Sigma^{-1/2}\xi\|_2 \le \alpha\}$ where $\Sigma$ satisfies $\Sigma^{-1}e \in \mathbb{R}^n_+$. Let $\{x_k\}$ be the proximal point sequence w.r.t. $D_\varphi(x, y) = \langle x - y, \Sigma(x - y) \rangle$, associated with sequence $\{\lambda_k\}$ and starting point $x_{\mathrm{R}} = \arg\min_{x \in \Delta^n} \max_{\xi \in \Xi(\infty)} \langle a_0 + \xi, x \rangle$. If the sequence $\{\omega_k\}$ is defined as*

$$\omega_k = \left(\sum_{j=0}^{k-1} \lambda_j^{-1}\right)^{-1}, \quad for\ k = 1, 2, ...,$$

*and let $\alpha(\omega_k)$ be such that $\arg\min_{x \in \Delta^n} \langle a_0, x \rangle + \alpha(\omega_k)\sqrt{\langle x, \Sigma x \rangle} = \arg\min_{x \in \Delta^n} \langle a_0, x \rangle + \omega_k \langle x, \Sigma x \rangle$. Then*

$$x_k = x_{\mathrm{PE}}(\alpha(\omega_k)), \quad for\ k = 1, 2, ....$$

We present the proof for Theorem 1 in Appendix A. In addition, we provide a closed-form solution for $\alpha(\omega_k)$ as a function of the current $\omega_k$ and the current PPM solution $x_k$ in Appendix E. Practically, after each PPM step, we know the current PPM solution $x_k$ is a robust solution with radius, $\alpha(\omega_k)$ which we can calculate in closed-form.

### 4.2 Implication of Theorem 1: PPM Iterates as Pareto Efficient Robust Solutions

Theorem 1 inspires an efficient algorithm for approximating multiple Pareto efficient robust solutions in two passes: first, solve the robust problem (RC) for $x_{\mathrm{R}}$; second, solve the nominal problem (P) with approximate PPM initialized with $x_{\mathrm{R}}$, finally the iterates of the approximate PPM are approximate Pareto efficient robust solutions. Specifically, we present the following algorithms.

---

**Algorithm 1** Multiple Approximate Pareto efficient Robust Solutions via Proximal Point Methods

---

**Input**: $\{\lambda_k\} \in \mathbb{R}_{++}$ satisfying $\sum_{k=0}^\infty \lambda_k^{-1} = +\infty$ **and** $\varphi : \mathcal{X} \to \mathbb{R}$ satisfying Assumption A1 and A2.
**Solve for** $x_{\mathrm{R}} = \arg\min_{x \in \mathcal{X}} \max_{a \in \mathcal{U}} f(x, a)$ **and set** $x_0 = x_{\mathrm{R}}$.
**for** $k = 0, 1, ...$ **do**
  $x_{k+1} \approx \arg\min_{x \in \mathcal{X}} f(x, a_0) + \lambda_k D_\varphi(x, x_k)$
**end for**
**return** $\{x_k\}$ as approximate efficiency-robustness Pareto efficient robust solutions.

---

Compared with the cost of the existing method: $N \times T_{\mathrm{RC}}$, the computation cost of algorithm 1 is $T_{\mathrm{RC}} + (N - 1) \times T_{\widehat{\mathrm{PPM}}}$, where $N$ is the number of robust solutions to be generated, $T_{\mathrm{RC}}$ is the cost of solving a single robust optimization problem, and $T_{\widehat{\mathrm{PPM}}}$ is the cost of a single step of an approximate PPM. In general, performing an exact proximal point method update is no easier than solving the robust optimization problem, therefore, the computation cost reduction is enjoyed only when we can equip Algorithm 1 with cheap approximate PPM. Specifically, for linear objective functions, i.e., $f(x, a) = \langle a, x \rangle$, the exact proximal point method updates are equivalent to projected gradient descent (PGD) updates with the same cost as solving the robust problem. For nonlinear differentiable objectives, the proximal point method can be approximated by computationally cheap first-order approximates such as gradient descent, extra-gradient method, and optimistic gradient method (Mokhtari et al. (2020a); Parikh et al. (2014)). In Section 5, we first validate our theoretical results using exact PPM under linear objective functions; we then demonstrate the computation cost reduction by Algorithm 1 equipped with approximate PPMs under nonlinear differentiable objective functions, where the conditions for our exact results in Theorem 1 no longer hold.

### 4.3 Probabilistic Performance Bound: Random Polyhedron Domain and Ellipsoidal Uncertainty Set

To extend the result to more general domains beyond simplex domains. In this subsection, we show that under ellipsoidal uncertainty sets, the performance (measured by efficiency and robustness) of

the Pareto efficient robust solutions with random polyhedron domains with constraint coefficients generated i.i.d. from bounded distributions are bounded between the performance of two sets of Pareto efficient robust solutions with two simplex domains with a small scaling factor, hence by Theorem 1, between the performance of two proximal point trajectories with high probability.

We denote the Pareto efficient robust solution with domain $\mathcal{S}$ and ellipsoidal uncertainty set $\Xi(\alpha) = \left\{ \xi \in \mathbb{R}^n : \|\Sigma^{-1/2}\xi\|_2 \leq \alpha \right\}$ as

$$x_{\mathrm{PE}}(\alpha, \mathcal{S}) = \arg \min_{x \in \mathcal{S}} \max_{\xi \in \Xi(\alpha)} \langle a_0 + \xi, x \rangle,$$

$$= \arg \min_{x \in \mathcal{S}} \langle a_0, x \rangle + \alpha \sqrt{\langle x, \Sigma x \rangle}.$$

Alternatively, we define

$$x'_{\mathrm{PE}}(\upsilon(\alpha), \mathcal{S}) = \arg \min \left\{ \sqrt{\langle x, \Sigma x \rangle} : \langle a_0, x \rangle \leq \upsilon(\alpha), \ x \in \mathcal{S} \right\},$$

where we assume $\upsilon(\alpha)$ is chosen such that $x'_{\mathrm{PE}}(\upsilon(\alpha), \mathcal{S}) = x_{\mathrm{PE}}(\alpha, \mathcal{S})$.

**Corollary 1.** *Consider a random polyhedron $\tilde{\mathcal{X}} = \{x \in \mathbb{R}^n_+ : \tilde{\mathbf{A}}x \leq \overline{d} \cdot e\}$, where $\tilde{\mathbf{A}} \in \mathbb{R}^{m \times n}$ and $\tilde{A}_{ij}$ are i.i.d. according to a bounded distribution with support $[0, b]$ and $\mathbb{E}[\tilde{A}_{ij}] = \mu$ for all $i \in [m], j \in [n]$. Define simplex $\Delta = \{x \in \mathbb{R}^n_+ : \langle e, x \rangle \leq \frac{\overline{d}}{b}\}$. Then for all $\alpha \in (0, \infty)$,*

$$\mathbb{P}\left( \mathrm{R}(x_{\mathrm{PE}}(\alpha, \Delta)) \leq \mathrm{R}(x_{\mathrm{PE}}(\alpha, \tilde{\mathcal{X}})) \leq \mathrm{R}(x_{\mathrm{PE}}(\alpha, \frac{b}{\mu(1-\epsilon)} \cdot \Delta)) \right) = 1 - \frac{1}{m},$$

*where $\epsilon = \frac{b}{\mu}\sqrt{\frac{\log m}{n}}$.*

The proof of Corollary 1 is presented in Appendix C.

## 4.4 MULTIPLE UNCERTAIN CONSTRAINTS

In this subsection, we extend the result to robust linear optimization problems with multiple uncertain linear constraints. We show that under uncertain linear constraints, the Pareto efficient robust solutions can be obtained by solving a set of saddle point problems. Specifically, we study the following problem

$$\text{(RCWUC)} \quad \min \left\{ \langle a_0, x \rangle : \sup_{\xi_i \in \Xi_i} \langle a_i + \xi_i, x \rangle - b_i \leq 0, \ \forall i \in [m], \ x \in \Delta^n \right\},$$

where $\Xi_i$ is the uncertainty set for uncertain parameter $\xi_i$ in constraint $i$ for all $i \in [m]$. We assume the constraints share the same ellipsoidal uncertainty set, i.e., $\Xi_i(r) = \{\xi \in \mathbb{R}^n : \|\Sigma^{-1/2}\xi\|_2 \leq r\}$, $\forall i \in [m]$. Consequently, the Pareto efficient robust solutions are the set

$$\{x'_{\mathrm{PE}}(\beta) : \ \beta \in [0, \infty)\},$$

where

$$x'_{\mathrm{PE}}(\beta) = \arg \min \left\{ \langle c_0, x \rangle : \sup_{\xi_i \in \Xi_i(\beta)} \langle c_i + \xi_i, x \rangle - b_i \leq 0, \ \forall i \in [m], \ x \in \Delta^n \right\}.$$

In general, the conventional approach for solving for a Pareto efficient robust solution requires dual reformulation of the robust constraints resulting in a computationally harder problem than the deterministic counterpart. Our next result shows for the specific problem setting of (RCWUC), its Pareto efficient robust solutions can be obtained by solving a set of saddle-point problems via gradient descent-ascent-like methods.

**Proposition 2.** $\{x'_{\mathrm{PE}}(\beta) : \ \beta \in [0, \infty)\}$ *is equal to a set of saddle-point solutions* $\{x_{\mathrm{SP}}(\alpha) : \ \alpha \in [0, \infty)\}$ *where*

$$x_{\mathrm{SP}}(\alpha) = \arg \min_{x \in \Delta^n} \left\{ \max_{\lambda \in \mathbb{R}^m_+} \langle c_0, x \rangle + \langle \lambda, Cx \rangle + \alpha \langle \lambda, e \rangle \langle x, \Sigma x \rangle - \langle \lambda, b \rangle \right\}.$$

The proof of Proposition 2 is presented in Appendix C. As a result, a set of approximate Pareto efficient robust solutions can be obtained by running the following algorithm for a set of $\alpha$ values.

---

**Algorithm 2** $\hat{x}_{\mathrm{SP}}(\alpha)$ Oracle

---

**input:** $\alpha, x_0 = \mathbf{0}, \lambda_0 = \mathbf{0}$
**for** $k = 0, 1, \cdots, T$ **do**
$\quad \lambda_{k+1} \leftarrow \arg\max_{\lambda \in \mathbb{R}^m_+} \langle \lambda, C x_k \rangle + \alpha \langle \lambda, e \rangle \langle x_k, \Sigma x_k \rangle - \langle \lambda, b \rangle$
$\quad x_{k+1} \leftarrow \arg\min_{x \in \Delta^n} \langle c_0 + \lambda_k^\top C, x \rangle + \alpha \langle \lambda_k, e \rangle \langle x, \Sigma x \rangle$
**end for**
**return** $\hat{x}_{\mathrm{SP}}(\alpha) = x_T$

---

## 5 Performance of Algorithm 1: Numerical Studies

In this section, we present two experiments testing the empirical performance of generating a set of approximate efficiency-robustness Pareto efficient robust solutions via Algorithm 1. The code for all the experiments is included in the supplementary material.

### 5.1 Robust Portfolio Optimization

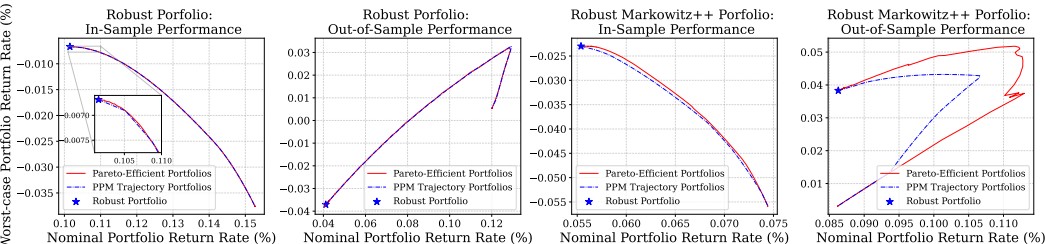

Figure 1: **Robust Portfolio Optimization:** The first two figures compare the in-sample and the out-of-sample performances (as measured by the nominal and worst-case returns) of the exact Pareto efficient portfolios against our PPM trajectory approximated Pareto efficient portfolios for the basic robust portfolio problem (RPO); the last two figures are from the same experiment but on the Markowitz++ problem. The results show Algorithm 1 generates closed approximations of Pareto efficient portfolios measured by both the in-sample and out-of-sample performances.

**Robust Portfolio Optimization.** In portfolio optimization, we are concerned with constructing a portfolio as a convex combination of $n$ stocks. The returns of the $n$ stocks are modeled by a random vector, $r$. We assume that from historical data, we estimated the expectation and the covariance matrix of $r$ to be $\mu$ and $\Sigma$. In the robust approach to the problem, we assume the realization of the uncertain return lies within an uncertainty set which we design as an ellipsoidal uncertainty set $\mathcal{U}(\alpha) = \{\mu + \xi \in \mathbb{R}^n : \|\Sigma^{-1/2}\xi\|_2 \le \alpha\}$. Our objective is to choose the portfolio weight that minimizes the worst-case loss (or equivalently maximizes the worst-case return) under a simplex domain, leading to the following robust portfolio optimization problem,

$$\text{(RPO)} \quad \min_{x \in \Delta^n} \max_{r \in \mathcal{U}(\alpha)} -\langle r, x \rangle.$$

As a consequence of Lemma 1, (RPO) is equivalent to the classic mean-standard deviation risk measure minimization problem,

$$\text{(RMM)} \quad \min_{x \in \Delta^n} -\langle \mu, x \rangle + \alpha \sqrt{\langle x, \Sigma x \rangle}.$$

To further test the performance of algorithm 1 beyond the simplex domain required for our exact result in Theorem 1, we also consider the extended Markowitz model (Markowitz++) proposed by Boyd et al. (2024) with additional practical constraints and objective terms corresponding to e.g., shorting, weight limits, cash holding/borrowing constraints and costs. The detailed formulation of the Markowitz++ portfolio problem is in Appendix D.

**Experiment design.** We first construct portfolios with in-sample historical stock return data, before testing the nominal and worst-case returns of each portfolio on out-of-sample stock return data. As the benchmark, we first construct exact Pareto efficient robust portfolios by solving exactly the

problem (RPO)/Markowitz++ under various levels of $\alpha$. Then we deploy algorithm 1 to generate approximate Pareto efficient robust portfolios, i.e., we use the (RPO)/Markowitz++ portfolio with the largest $\alpha$ to initialize the PPM method for solving the nominal portfolio optimization problem with $\alpha = 0$, the PPM trajectory are approximate Pareto efficient robust portfolios. Specifically, we choose our decision space as 20 stocks from S&P 500 companies. We use the historical daily return data of the 20 stocks over 3 years (from 2021-01-01 to 2023-12-30) and estimate the moment information $\mu_{\text{in}}$ and $\Sigma_{\text{in}}$ as our in-sample data for constructing portfolios; the assumed unseen future daily return of the 20 stock for the next 8 months (from 2024-01-01 to 2024-08-01) are used to estimate $\mu_{\text{out}}$ and $\Sigma_{\text{out}}$ which serve as our out-of-sample data for evaluating the nominal and worst-case returns of each portfolio.

**Results.** The result is shown in figure 1. For the vanilla robust portfolio problem (RPO), although the assumption $\Sigma^{-1}e \in \mathbb{R}^n_+$ in Theorem 1 is not satisfied in this experiment, the performance of PPM method generated portfolios matches closely to that of the exact Pareto efficient robust portfolios. For the Markowitz++ portfolio problem, the domain further deviates from our simplex requirement for the exact result in Theorem 1, algorithm 1 still produces good approximate Pareto efficient robust portfolios. The Markowitz++ out-of-sample performance differences are attributed to two factors: first, despite similar in-sample performances, the numerical stock weights between the two portfolios under each $\alpha$ have absolute differences up to $10\%$ for some stocks; second, the in-sample and out-of-sample stock return distributions shifted significantly, with some elements of the expected returns and the covariance matrix underwent sign changes.

As we discuss in section 4.2, the robust portfolio problem with a linear objective does not enjoy computation reduction by using the PPM approach. We also point out closed-form solution exists for the standard Markowitz Portfolio but not for the Markowitz++ Portfolio. In the next experiment, we demonstrate for nonlinear differentiable objectives, the PPM approach equipped with cheap first-order approximate PPM provides considerable computation cost reduction compared with the traditional approach. Specifically, we deploy the PPM approach on adversarially robust deep learning with nonconvex-nonconcave objective functions.

## 5.2 ADVERSARIALLY ROBUST DEEP LEARNING

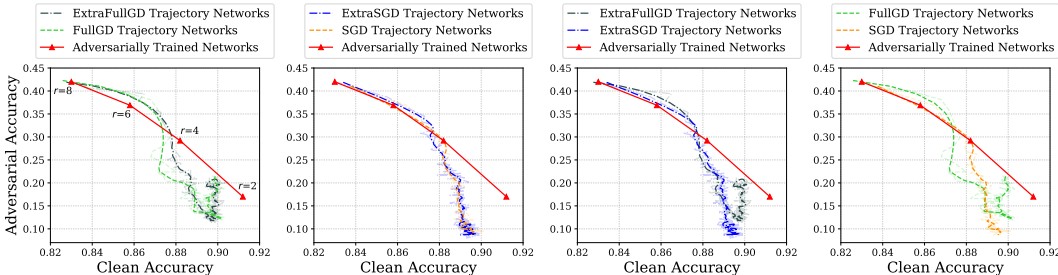

Figure 2: **Adversarially Robust Deep Learning**: Clean test accuracy and PGD adversarial test accuracy of Algorithm 1 generated approximate Pareto efficient robust networks v.s. adversarially trained Pareto efficient robust networks. Four variants of the gradient method are implemented in Algorithm 1 for standard training with robust parameter initialization. The first two figures show performance improvement by using the extra gradient, the last two figures show performance improvement by using the full gradient. One insight is that the performance of Algorithm 1 can be improved by improving the gradient method approximation to PPM during standard training. The best variant: Algorithm 1 with ExtraFullGD generated 100 approximations of Pareto efficient robust networks in two algorithmic passes.

**Adversarial training as robust optimization.** The goal in adversarially robust deep learning is to learn networks that are robust against adversarial attacks (i.e., perturbations on the input examples that aim to deteriorate the accuracy of classifiers). A common strategy to robustify networks is adversarial training, which can be formulated as the following robust optimization problem,

$$\min_{\theta} \mathbb{E}_{(x,y)\sim\mathcal{D}} \left[ \max_{\xi \in \Xi} \ell(f_\theta(x + \xi), y) \right],\tag{8}$$

Table 1: Computation Cost: Algorithm 1 vs. Adversarial Training

| Method | Cost per Pareto efficient robust network (min) | Cost to generate $N$ Pareto efficient robust networks (min) |
|---|---|---|
| Algorithm 1 with ExtraFullGD | $0.25\ (= T_{\widetilde{\text{PPM}}})$ | $15.12 + 0.25(N-1)$ |
| FGSM (Wong et al. (2020)) | $15.12\ (= T_{\text{RC}})$ | $15.12N$ |

where $\mathcal{D}$ is the distribution generating pairs of examples $x \in \mathbb{R}^d$ and labels $y \in [c]$, $f_\theta$ is a neural network parameterized by $\theta$, $\xi$ is the perturbation with perturbation set $\Xi$, and $\ell$ is the lost function. Typically, the perturbation set is designed as norm-balls, $\mathbb{B}_{\|\cdot\|}(r)$ whose size can be controlled via the radius parameter, $r$. Naturally, we can apply our framework and define a clean accuracy-adversarial accuracy Pareto efficient robust networks as networks adversarially trained under different levels of perturbation set radius, $r$. The trade-off between clean accuracy and adversarial accuracy in adversarially robust deep learning has been observed empirically Wang et al. (2020); Su et al. (2018) and studied theoretically Raghunathan et al. (2020); Tsipras et al. (2019); Pang et al. (2022). In practice, given the nonconvex-nonconcave loss function, the adversarial training is solved approximately via iteratively performing the following: first approximately solving the inner maximization problem, followed by gradient method update on the parameter $\theta$ (Madry et al. (2018)).

**Experiment design.** As the benchmark for algorithm 1, we first adversarially train networks under different perturbation set radius, $r$ to learn a set of clean accuracy-adversarial accuracy Pareto efficient robust networks. Then we run Algorithm 1 by performing standard training but with the key difference of initializing the network parameter with the parameter of the most robust network (i.e., the adversarially trained network with the largest $r$). Finally, the output of Algorithm 1, i.e., the standard training parameter sequence with robust parameter initialization, $\{\theta_k\}$ corresponds to a set of approximate clean accuracy-adversarial accuracy Pareto efficient robust networks. We use four variants of approximate PPMs (i.e., stochastic/full vanilla/Extra gradient method). The detailed experiment setup is presented in Appendix F.

**Solution Quality.** The results evaluating the performance of our PPM-based procedure for generating approximate clean (test) accuracy-adversarial (test) accuracy Pareto efficient robust networks are shown in Figure 2. The networks generated by our gradient method trajectories initially approximate/surpass both the clean and adversarial accuracy of adversarially trained networks with perturbation set radius, $r$ in $\{8, 6, 4\}$, before generating networks with both lower clean and adversarial accuracy that than of adversarially trained network with $r = 2$. The initial approximation/surpassing in performance against adversarially trained networks shows our procedure can generate adversarially robust networks as effective as traditional adversarial training. The later drop in performance is mainly contributed by the low learning rate, although tuning the gradient methods is not the focus of our paper, it opens up future works for investigating the gradient method designs in Algorithm 1 that can improve upon our current approximates using constant learning rates. In Appendix F.1, we show this later drop in performance can be overcome by an additional run of algorithm 1, this time initialized with the adversarially trained network with $r = 4$. Comparing the performance across the four trajectories corresponding to the four variants of the approximate PPMs, the ExtraFullGD trajectory generated networks have the best performance. This result is as expected since ExtraFullGD is the best approximation for PPM.

**Computation Cost.** The computation cost result in Table 1 shows our PPM-based procedure reduces the computation cost of generating $N$ approximate Pareto efficient robust networks from $15.12 \times N$ to $15.12 + 0.25 \times (N-1)$, where $T_{\text{RC}} = 15.12$ is the cost of a single adversarial training; $T_{\widetilde{\text{PPM}}} = 0.25$ is the cost per approximate PPM (ExtraFullGD) iterate. Specifically, algorithm 1 generates a set of $N$ approximate Pareto efficient robust networks with the cost of a single pass of adversarial training (costing 15.12 minutes) , plus an algorithmic pass of standard training with robust parameter initialization for $N - 1$ epochs, where each epoch of standard training costs 0.25 minutes, and one approximate Pareto efficient robust network is generated per epoch.

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

# A    PROOF OF THEOREM 1

Theorem 1 follows by the combination of Proposition 1, 3 and 4, which together show the notions of proximal-point sequences, central path, mean-variance risk minimization solutions and Pareto efficient robust solutions all coincide under simplex domain and ellipsoidal uncertainty set. Here we present Proposition 3 and 4.

**Proposition 3** (**Correspondence between Mean-Variance Risk Minimization Solutions and the Central Path**). *Under simplex domain* $\Delta^n := \left\{ x \in \mathbb{R}^n_+ : \langle e, x \rangle = 1 \right\}$, *assume the covariance matrix of the returns* $\Sigma$ *satisfies* $\Sigma^{-1} e \in \mathbb{R}^n_+$, *Let the set of Pareto efficient solutions to the mean-variance risk measure minimization problem be*

$$\left\{ x_{\mathrm{RM}}(\alpha) = \arg \min \left\{ \langle a_0, x \rangle + \alpha \langle x, \Sigma x \rangle : x \in \Delta^n \right\} : \alpha \in (0, \infty) \right\}.$$

*Let the central path be*

$$\left\{ x_{\mathrm{CP}}(\omega) = \arg \min \left\{ \langle a_0, x \rangle + \omega D_{\varphi, \Sigma}(x, x_{\mathrm{mv}}) : x \in \Delta^n \right\} : \omega \in (0, \infty) \right\},$$

*where* $D_{\varphi, \Sigma}(x, x_{\mathrm{mv}}) = \langle x - x_{\mathrm{mv}}, \Sigma(x - x_{\mathrm{mv}}) \rangle$ *which is induced by* $\varphi(x) = \langle x, \Sigma x \rangle$ *and* $x_{\mathrm{mv}}$ *is the minimum variance solution defined as* $x_{\mathrm{mv}} = \arg \min \left\{ \langle x, \Sigma x \rangle : x \in \Delta^n \right\}$. *Then*

$$x_{\mathrm{RM}}(\alpha) = x_{\mathrm{CP}}(\alpha), \quad \forall \alpha \in (0, \infty).$$

The proof of Proposition 3 is presented in Appendix C.

**Proposition 4** (**Correspondence Between Pareto-Efficiency Robust Solutions and the Central Path**). *Under simplex domain* $\Delta^n = \left\{ x \in \mathbb{R}^n_+ : \langle e, x \rangle = 1 \right\}$ *and ellipsoidal uncertainty set* $\Xi(\alpha) = \left\{ \xi \in \mathbb{R}^n : \| \Sigma^{-1/2} \xi \|_2 \leq \alpha \right\}$ *where* $\Sigma$ *satisfies* $\Sigma^{-1} e \in \mathbb{R}^n_+$, *let the Pareto efficient robust solutions be*

$$\left\{ x_{\mathrm{PE}}(\alpha) = \arg \min_{x \in \Delta^n} \max_{\xi \in \Xi(\alpha)} \langle a_0 + \xi, x \rangle : \alpha \in (0, \infty) \right\}.$$

*Let the central path be*

$$\left\{ x_{\mathrm{CP}}(\omega) = \arg \min_{x \in \Delta^n} \langle a_0, x \rangle + \omega D_{\varphi, \Sigma}(x, x_{\mathrm{R}}) : \omega \in (0, \infty) \right\}$$

*where* $D_{\varphi, \Sigma}(x, x_{\mathrm{R}}) = \langle x - x_{\mathrm{R}}, \Sigma(x - x_{\mathrm{R}}) \rangle$ *which is induced by* $\varphi(x) = \langle x, \Sigma x \rangle$ *and* $x_{\mathrm{R}} = \arg \min_{x \in \Delta^n} \max_{\xi \in \Xi(\infty)} \langle a_0 + \xi, x \rangle$. *Let* $\alpha(\omega)$ *be such that* $\arg \min_{x \in \Delta^n} \langle a_0, x \rangle + \alpha(\omega) \sqrt{\langle x, \Sigma x \rangle} = \arg \min_{x \in \Delta^n} \langle a_0, x \rangle + \omega \langle x, \Sigma x \rangle$. *Then*

$$x_{\mathrm{PE}}(\alpha(\omega)) = x_{\mathrm{CP}}(\omega), \quad \forall \alpha \in (0, \infty).$$

The proof of Proposition 4 is presented in Appendix C.

# B    PROOFS OF RESULTS IN SECTION 3

**Proof of Lemma 1.** Denote $z = \Sigma^{-1/2} \xi$,

$$\min_{x \in \mathcal{X}} \max_{\xi \in \Xi(\alpha)} \langle a_0 + \xi, x \rangle$$

$$= \min_{x \in \mathcal{X}} \langle a_0, x \rangle + \max_{\xi \in \Xi(\alpha)} \langle \xi, x \rangle$$

$$= \min_{x \in \mathcal{X}} \langle a_0, x \rangle + \max_{z \in \mathbb{B}_{\| \cdot \|_2}(\alpha)} \langle z, \Sigma^{1/2} x \rangle$$

$$= \min_{x \in \mathcal{X}} \langle a_0, x \rangle + \alpha \sqrt{\langle x, \Sigma x \rangle}.$$

## C   PROOFS OF RESULTS IN SECTION 4

**Lemma 2.** *Assume* $\Sigma^{-1}e \in \mathbb{R}_+^n$,

$$\arg\min\{\langle x, \Sigma x\rangle : \ x \in \Delta^n\} = \frac{\Sigma^{-1}e}{\langle e, \Sigma^{-1}e\rangle}.$$

**Proof.** By KKT condition,

$$\arg\min\{\langle x, \Sigma x\rangle : \ \langle e, x\rangle = 1\} = \frac{\Sigma^{-1}e}{\langle e, \Sigma^{-1}e\rangle},$$

denote $\overline{x} = \frac{\Sigma^{-1}e}{\langle e, \Sigma^{-1}e\rangle}$ we have,

$$\langle \overline{x}, \Sigma\overline{x}\rangle \le \langle x, \Sigma x\rangle, \quad \forall x \in \{x \in \mathbb{R}^n : \langle e, x\rangle = 1\}.$$

Given $\Delta^n := \{x \in \mathbb{R}_+^n : \langle e, x\rangle = 1\} \subset \{x \in \mathbb{R}^n : \langle e, x\rangle = 1\}$, we have

$$\langle \overline{x}, \Sigma\overline{x}\rangle \le \langle x, \Sigma x\rangle, \quad \forall x \in \Delta^n.$$

Finally, by $\Sigma^{-1}e \in \mathbb{R}_+^n$,

$$\overline{x} \in \Delta^n,$$

hence, $\arg\min\{\langle x, \Sigma x\rangle : \ x \in \Delta^n\} = \frac{\Sigma^{-1}e}{\langle e, \Sigma^{-1}e\rangle}$.

**Proof of Proposition 3.** For all $\alpha \in (0, \infty)$ we have

$$\begin{aligned}
x_{\mathrm{CP}}(\alpha) &= \arg\min\{\langle a_0, x\rangle + \alpha D_{\varphi, \Sigma}(x, x_{\mathrm{mv}}) : \ x \in \Delta^n\}\\
&= \arg\min\{\langle a_0, x\rangle + \alpha\langle x - x_{\mathrm{mv}}, \Sigma(x - x_{\mathrm{mv}})\rangle : \ x \in \Delta^n\}\\
&= \arg\min\{\langle a_0, x\rangle + \alpha\langle x, \Sigma x\rangle - 2\alpha\langle x_{\mathrm{mv}}, \Sigma x\rangle : \ x \in \Delta^n\}\\
&= \arg\min\left\{\langle a_0, x\rangle + \alpha\langle x, \Sigma x\rangle - \frac{2\alpha}{\langle e, \Sigma^{-1}e\rangle}\langle e, x\rangle : \ x \in \Delta^n\right\}\\
&= \arg\min\left\{\langle a_0, x\rangle + \alpha\langle x, \Sigma x\rangle - \frac{2\alpha}{\langle e, \Sigma^{-1}e\rangle} : \ x \in \Delta^n\right\}\\
&= \arg\min\{\langle a_0, x\rangle + \alpha\langle x, \Sigma x\rangle : \ x \in \Delta^n\}\\
&= x_{\mathrm{RM}}(\alpha),
\end{aligned}$$

where the fourth equality is by Lemma 2, and the fifth equality is due to $x \in \Delta^n$.

**Proof of Proposition 4.** The result is a direct consequence of Lemma 1 and Proposition 3.

$$\begin{aligned}
x_{\mathrm{PE}}(\alpha(\omega)) &= \arg\min_{x \in \Delta^n}\max_{\xi \in \Xi(\alpha(\omega))}\langle a_0 + \xi, x\rangle\\
&= \arg\min_{x \in \Delta^n}\langle a_0, x\rangle + \alpha(\omega)\sqrt{\langle x, \Sigma x\rangle}\\
&= \arg\min_{x \in \Delta^n}\langle a_0, x\rangle + \omega\langle x, \Sigma x\rangle\\
&= \arg\min_{x \in \Delta^n}\langle a_0, x\rangle + \omega D_{\varphi, \Sigma}(x, x_{\mathrm{MV}})\\
&= \arg\min_{x \in \Delta^n}\langle a_0, x\rangle + \omega D_{\varphi, \Sigma}(x, x_{\mathrm{R}})\\
&= x_{\mathrm{CP}}(\omega).
\end{aligned}$$

The second equality is due to Lemma 1, the fourth equality is by Proposition 3, and finally the fifth equality follows from Lemma 1.

Our performance bound in corollary 1. builds on the following result that a random polyhedron with constraint coefficients generated i.i.d. according to bounded distributions is sandwiched between two simplices with high probability.

**Lemma 3** (**Theorem 2.1. in El Housni & Goyal (2017)**). *Consider a random polyhedron $\tilde{\mathcal{X}} = \{x \in \mathbb{R}^n_+ : \tilde{\mathbf{A}}x \leq \bar{d} \cdot e\}$, where $\tilde{\mathbf{A}} \in \mathbb{R}^{m \times n}$ and $\tilde{A}_{ij}$ are i.i.d. according to a bounded distribution with support $[0, b]$ and $\mathbb{E}[\tilde{A}_{ij}] = \mu$ for all $i \in [m], j \in [n]$. Define simplex $\Delta = \{x \in \mathbb{R}^n_+ : \langle e, x \rangle \leq \frac{\bar{d}}{b}\}$. Then*

$$\mathbb{P}\left(\Delta \subseteq \tilde{\mathcal{X}} \subseteq \frac{b}{\mu(1-\epsilon)} \cdot \Delta\right) = 1 - \frac{1}{m},$$

*where $\epsilon = \frac{b}{\mu}\sqrt{\frac{\log m}{n}}$.*

**Proof of Corollary 1.** Denote $\kappa = \frac{b}{\mu(1-\epsilon)}$ where $\epsilon = \frac{b}{\mu}\sqrt{\frac{\log m}{n}}$.

$$\mathbb{P}\left(\Delta \subseteq \tilde{\mathcal{X}} \subseteq \kappa \cdot \Delta\right)$$

$$=\mathbb{P}\left(\min_{x:\langle a_0, x\rangle \leq \upsilon(\alpha),\ x \in \kappa \cdot \Delta} \sqrt{\langle x, \Sigma x\rangle} \leq \min_{x:\langle a_0, x\rangle \leq \upsilon(\alpha),\ x \in \tilde{\mathcal{X}}} \sqrt{\langle x, \Sigma x\rangle} \leq \min_{x:\langle a_0, x\rangle \leq \upsilon(\alpha),\ x \in \Delta} \sqrt{\langle x, \Sigma x\rangle}\right)$$

$$=\mathbb{P}\left(\sqrt{\langle x, \Sigma x\rangle}|x = x'_{\text{PE}}(\upsilon(\alpha), \kappa \cdot \Delta) \leq \sqrt{\langle x, \Sigma x\rangle}|x = x'_{\text{PE}}(\upsilon(\alpha), \tilde{\mathcal{X}}) \leq \sqrt{\langle x, \Sigma x\rangle}|x = x'_{\text{PE}}(\upsilon(\alpha), \cdot \Delta)\right)$$

$$=\mathbb{P}\left(\text{R}(x'_{\text{PE}}(\upsilon(\alpha), \Delta)) \leq \text{R}(x'_{\text{PE}}(\upsilon(\alpha), \tilde{\mathcal{X}})) \leq \text{R}(x'_{\text{PE}}(\upsilon(\alpha), \kappa \cdot \Delta))\right)$$

$$=\mathbb{P}\left(\text{R}(x_{\text{PE}}(\alpha, \Delta)) \leq \text{R}(x_{\text{PE}}(\alpha, \tilde{\mathcal{X}})) \leq \text{R}(x_{\text{PE}}(\alpha, \kappa \cdot \Delta))\right).$$

By Lemma 3 we have result.

**Proof of Proposition 2.** By Proposition 1,

$$x'_{\text{PE}}(\beta) = \arg\min\left\{\langle c_0, x\rangle : \langle c_i, x\rangle + \beta\sqrt{\langle x, \Sigma x\rangle} - b_i \leq 0,\ \forall i \in [m],\ x \in \Delta^n\right\}.$$

By the monotonicity of $\sqrt{\cdot}$,

$$\{x_{\text{SP}}(\alpha):\ \alpha \in [0, \infty)\} = \{x'_{\text{PE}}(\beta):\ \beta \in [0, \infty)\},$$

where

$$x_{\text{SP}}(\alpha) = \arg\min\left\{\langle c_0, x\rangle : \langle c_i, x\rangle + \alpha\langle x, \Sigma x\rangle - b_i \leq 0,\ \forall i \in [m],\ x \in \Delta^n\right\}.$$

Applying the method of Lagrangian, $x_{\text{SP}}(\alpha)$ can reformulated as the solution to the following saddle-point problem

$$x_{\text{SP}}(\alpha) = \arg\min_{x \in \Delta^n}\left\{\max_{\lambda \in \mathbb{R}^m_+} \langle c_0, x\rangle + \sum_{i \in [m]} \lambda_i \left(\langle c_i, x\rangle + \alpha\langle x, \Sigma x\rangle - b_i\right)\right\}$$

$$= \arg\min_{x \in \Delta^n}\left\{\max_{\lambda \in \mathbb{R}^m_+} \langle c_0, x\rangle + \langle \lambda, Cx\rangle + \alpha\langle \lambda, e\rangle\langle x, \Sigma x\rangle - \langle \lambda, b\rangle\right\}.$$

# D    MARKOWITZ++

The Markowitz++ proposed by Boyd et al. (2024) extends the classical Markowitz Portfolio by introducing practical constraints and objective terms. In the experiment, we include the option for shorting, holding limits on each asset, the option for cash holding/borrowing, and the associated costs. The formulation of the Markowitz++ problem is shown below,

$$\min \quad -\langle \mu, x\rangle + \gamma^{\text{hold}}\phi^{\text{hold}}(x, c) + \alpha\sqrt{\langle x, \Sigma x\rangle} \tag{14a}$$

$$\text{s.t.} \quad \langle e, x\rangle + c = 1, \tag{14b}$$

$$x^{\text{min}} \leq x \leq x^{\text{max}}, \tag{14c}$$

$$c^{\text{min}} \leq c \leq c^{\text{max}}. \tag{14d}$$

The decision variables $x \in \mathbb{R}^n$ are stock weights ($x_i > 0$ for holding, $x_i < 0$ for shorting), $c \in \mathbb{R}$ is the cash weight ($c > 0$ denotes diluting the portfolio with cash, $c < 0$ denotes borrowing). The objective function now includes the additional term of holding costs, $\phi^{\text{hold}}(x, c)$ as defined by equation (5) in Boyd et al. (2024) and weighted by the parameter $\gamma^{\text{hold}}$. The first constraint enforces all weights sum to one, the second constraint corresponds to the maximum shorting position and maximum holding position on each asset, and the last constraint applies lower and upper limits on the cash weight. Finally, the nominal and worst-case return trade-off is controlled by adjusting the parameter, $\alpha$.

## E   CLOSED-FORM SOLUTION TO $\alpha(\omega_k)$

In Theorem 1, we wish to find $\alpha(\omega_k)$ such that $\arg\min_{x \in \Delta^n} \langle a_0, x \rangle + \alpha(\omega_k)\sqrt{\langle x, \Sigma x \rangle} = \arg\min_{x \in \Delta^n} \langle a_0, x \rangle + \omega_k \langle x, \Sigma x \rangle$. The following result provide a closed-form solution to $\alpha(\omega_k) = \alpha(\omega_k, x_k)$ as a function of $\omega_k$ and the current PPM solution $x_k$.

**Lemma 4.** *Under linear objective functions, $f(x, a) := \langle a, x \rangle$. Let $\{x_{\text{PE}}(\alpha) : \alpha > 0\}$ be the set of Pareto efficient robust solutions under simplex domain $\Delta^n = \{x \in \mathbb{R}^n_+ : \langle e, x \rangle = 1\}$ and ellipsoidal uncertainty set $\Xi(\alpha) = \{\xi \in \mathbb{R}^n : \|\Sigma^{-1/2}\xi\|_2 \leq \alpha\}$ where $\Sigma$ satisfies $\Sigma^{-1}e \in \mathbb{R}^n_+$. Let $\{x_k\}$ be the proximal point sequence w.r.t. $D_\varphi(x, y) = \langle x - y, \Sigma(x - y) \rangle$, associated with sequence $\{\lambda_k\}$ and starting point $x_{\text{R}} = \arg\min_{x \in \mathcal{X}} \max_{\xi \in \Xi(\infty)} \langle a_0 + \xi, x \rangle$. If the sequence $\{\omega_k\}$ is defined as*

$$\omega_k = \left( \sum_{j=0}^{k-1} \lambda_j^{-1} \right)^{-1}, \quad \text{for } k = 1, 2, ...,$$

*then for*

$$\alpha(\omega_k, x_k) = 2\omega_k \|\Sigma^{1/2} x_k\|_2,$$

*we have*

$$\arg\min_{x \in \Delta^n} \langle a_0, x \rangle + \alpha(\omega_k, x_k)\sqrt{\langle x, \Sigma x \rangle} = \arg\min_{x \in \Delta^n} \langle a_0, x \rangle + \omega_k \langle x, \Sigma x \rangle$$

*and*

$$x_k = x_{\text{PE}}(\alpha(\omega_k, x_k)), \quad \text{for } k = 1, 2, ....$$

**Proof.** First, by proposition 3 and 1 we have

$$\arg\min_{x \in \Delta^n} \langle a_0, x \rangle + \omega_k \langle x, \Sigma x \rangle = x_{\text{RM}}(\omega_k)$$

$$= x_{\text{CP}}(\omega_k)$$

$$= x_k.$$

Given

$$\arg\min_{x \in \Delta^n} \langle a_0, x \rangle + \omega_k \langle x, \Sigma x \rangle = x_k,$$

by KKT condition, we have $(x_k, \lambda^*, \mu^*)$ such that

$$\begin{cases} a_0 + 2\omega_k \Sigma x_k + \lambda^* e - \mu^* = 0 \\ -\langle \mu^*, x_k \rangle = 0, \ -x_k \leq 0, \ \mu^* \geq 0 \\ \langle e, x_k \rangle = 1. \end{cases} \tag{16}$$

Assume

$$\arg\min_{x \in \Delta^n} \langle a_0, x \rangle + \alpha\sqrt{\langle x, \Sigma x \rangle} = x^*,$$

by KKT condition, we have $(x^*, \kappa^*, \nu^*)$ such that

$$\begin{cases} a_0 + \alpha \dfrac{\Sigma x^*}{\|\Sigma^{-1/2} x^*\|_2} + \kappa^* e - \nu^* = 0 \\ -\langle \nu^*, x^* \rangle = 0, \ -x^* \leq 0, \ \nu^* \geq 0 \\ \langle e, x^* \rangle = 1. \end{cases} \tag{17}$$

If $\alpha = 2\omega_k\|\Sigma^{1/2}x_k\|_2$, (17) reduces to

$$
\begin{cases}
a_0 + 2\omega_k \dfrac{\|\Sigma^{-1/2}x_k\|_2}{\|\Sigma^{-1/2}x^*\|_2}\Sigma x^* + \kappa^* e - \nu^* = 0 \\
- \langle \nu^*, x^* \rangle = 0, \ -x^* \leq 0, \ \nu^* \geq 0 \\
\langle e, x^* \rangle = 1.
\end{cases}
\tag{18}
$$

By condition (16), $(x^* = x_k, \kappa^* = \lambda^*, \nu^* = \mu^*)$ also satisfy the KKT condition (18).

As a result, if $\alpha = 2\omega_k\|\Sigma^{1/2}x_k\|_2$, we have

$$
\arg\min_{x \in \Delta^n} \langle a_0, x \rangle + \alpha\sqrt{\langle x, \Sigma x \rangle} = \arg\min_{x \in \Delta^n} \langle a_0, x \rangle + \omega_k\langle x, \Sigma x \rangle = x_k
$$

and by Theorem 1,

$$
x_k = x_{\mathrm{PE}}(\alpha), \quad \text{for } k = 1, 2, \dots.
$$

# F  ADVERSARIALLY ROBUST DEEP LEARNING: EXPERIMENT SETUP

We use the CIFAR10 dataset, a PreAct ResNet18 architecture, and the cross entropy loss. The set of Pareto efficient robust networks are adversarially trained using fast gradient sign method (FGSM) with random initialization and fast training methods (Wong et al. (2020)) (shown to be as effective as projected gradient descent (PGD)-based training but with a much lower cost) and $l_\infty$ norm ball perturbation sets, $\mathbb{B}_{\|\cdot\|_\infty}(r)$, with $r$ in $\{2, 4, 6, 8\}$. The Algorithm 1 is initialized with the parameter of the adversarially trained network with perturbation sets $\mathbb{B}_{\|\cdot\|_\infty}(r = 8)$, after initialization, four variants of the standard training are performed with the following gradient methods: 1) Vanilla stochastic gradient descent (SGD); 2) Stochastic extra gradient descent (ExtraSGD); 3) Gradient descent with the gradient of the full train set (FullGD); 4) Extra gradient descent with the gradient of the full train set (ExtraFullGD). The implementation for extra gradient descent is adopted from Gidel et al. (2019). Each variant can be considered as an approximation to PPM in Algorithm 1, where ExtraSGD is a better approximation to PPM than SGD, and ExtraFullGD is a better approximation to PPM than ExtraSGD. SGD and ExtraSGD have a learning rate of $5\mathrm{e}-4$; FullGD and ExtraFullGD have a learning rate of $4\mathrm{e}-4$. The four variants of the standard training are each performed for 100 epochs, generating one approximate Pareto efficient robust network per epoch. The trajectories of four variants of the standard training are each a set of approximate clean accuracy-adversarial accuracy Pareto efficient robust networks. Finally, all networks' clean accuracy and adversarial accuracy are evaluated correspondingly on a clean test set and on an adversarial test set with PGD attacks.

## F.1   ALGORITHM 1 WITH ADDITIONAL INITIALIZATION

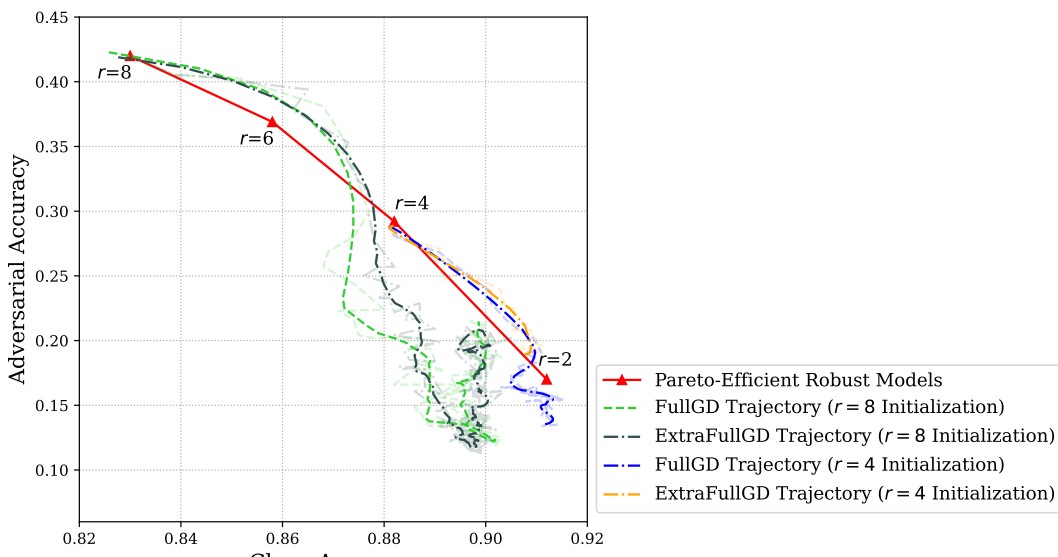

Figure 3: **Adversarially Robust Deep Learning: Algorithm 1 with Additional Initialization**:
Clean test accuracy and PGD adversarial test accuracy of Algorithm 1 generated approximate Pareto
efficient robust networks v.s. adversarially trained Pareto efficient robust networks. The red triangles
denote adversarially trained robust networks with perturbation ser radius, $r$ in $\{8, 6, 4, 2\}$. The black
and green lines denote the FullGD and ExtraFullGD generated robust models initialized with the
adversarially trained robust network with $r = 8$; The orange and blue lines denote the FullGD
and ExtraFullGD generated robust models initialized with the adversarially trained robust network
with $r = 4$. The main result is with the extra cost of one adversarial training and one standard
training, algorithm 1 generated models outperformance adversarially trained models across different
perturbation set radius values.

