# OpenReview forum: "Approximating Multiple Robust Optimization Solutions in One Pass via Proximal Point Methods"
_ICLR.cc/2025/Conference — Submitted to ICLR 2025_

### Official Review · Reviewer_7fPc · 2024-11-04

**Soundness:** 3
**Presentation:** 3
**Contribution:** 1
**Rating:** 5
**Confidence:** 4

**Summary:**

The paper proves that for robust LPs with uncertain objective functions under the simplex decision domain and ellipsoidal uncertainty sets, the proximal point trajectory are exactly Pareto efficient robust solutions. For robust LPs with a random polyhedron domain, the paper proves that with high probability, the performances of the Pareto efficient robust solutions are bounded by the performances of two proximal point trajectories. Numerical experiments on portfolio optimization and adversarially robust deep learning are provided.

**Strengths:**

The paper raises the interesting question of computing the efficient frontier that balances robustness and efficiency, and proposes a novel approach to finding this frontier. The paper provides an original proof that with high probability, the performances of the Pareto efficient robust solutions are bounded by the performances of two proximal point trajectories. The paper demonstrates clear structural organization. The visualizations are informative, and both the core ideas and theorems are presented with clarity.

**Weaknesses:**

The paper claims that the computational cost is reduced from N*T to 2*T. However, in each iteration of the proposed proximal point method for portfolio optimization, a quadratic optimization problem needs to be solved. Thus, the total computational cost is not reduced compared to the brute-force method of solving the problem for multiple alphas.

**Questions:**

The paper would benefit from numerical experiments comparing the running time of the proposed method against the brute-force approach. The abstract should explicitly state that mean-variance optimization is the paper's primary focus. Additionally, it would be helpful to acknowledge the existence of a closed-form solution to the Markowitz optimization problem. Regarding Figure 1, the substantial discrepancy between the out-of-sample performance of robust Markowitz++ portfolios, despite their similar in-sample performance, requires further explanation.

Typos:
Line 157 bracket is misplaced
Line 180 by -> be
Figure 1: Porfolio -> Portfolio

---

> ### Author Response · Authors · 2024-11-27
>
> We thank the reviewer for all the constructive comments. We aim to address all the comments here, please also refer to the `revised manuscript` with all major updates in blue.
>
> ### **[P1] Computation cost**:
>
> We thank the reviewer for raising this important point: i) The computation cost reduction is not enjoyed under linear objectives; it is enjoyed only under nonlinear objectives with approximate PPM.
>
>
>
> We agree it is important to discuss in more depth the exact condition under which the computational cost can be reduced and what is the fundamental trade-off.
>
>
>
> In short, the computational cost is reduced under the conditions that
>
>
>
> - the objective function is nonlinear and differentiable;
>
>
>
> - Subsequently allowing the use of cheap approximate PPM updates that generate approximate robust solutions.
>
>
>
> We expand on this point in the following.
>
>
>
> - More precisely, the cost of our method for generating N robust solutions is: $T_{\mathrm{R}} + (N-1)\cdot T_{\mathrm{\widetilde{PPM}}}$, where $T_{\mathrm{R}}$ is the cost of solving an instance of the robust optimization problem for initialization, and $T_{\mathrm{\widetilde{PPM}}}$ is the cost of a single iterate of approximate PPM. The cost of the brute-force method is $N\cdot T_{\mathrm{R}}$. Thus, the cost can be reduced only when $T_{\mathrm{\widetilde{PPM}}} < T_{\mathrm{R}}$.
>
>
>
> - For (constrained) linear problems, a first-order approximate PPM, i.e., (Projected) GD is equivalent to the exact PPM and thus has the same computation cost as solving the robust counterpart. For (constrained) problems with nonlinear and differentiable objectives, a first-order approximate PPM (e.g. (Projected) GD, MD, Extragradient) is significantly cheaper than an exact PPM, thus cheaper than solving an instance of the robust counterpart.
>
>
>
> - For (constrained) problems with nonlinear and differentiable objectives, the fundamental trade-off of our approach is computation cost v.s. robust solution quality. Our approach provides an operationalizable framework to adjust this trade-off, i.e., we can use better approximate PPM to generate better approximate robust solutions but at a higher cost.
>
>
>
>
>
>
>
> **Action taken**:
>
>
>
> We give a more extensive discussion on computational cost in `Section 4.2` of the revised manuscript, laying out the exact condition under which our approach provides computational cost reduction.
>
>
>
> We rectified the previous incorrect discussion of the computation cost for the portfolio optimization problem, and in the adversarial ML experiment, gave an extended discussion on the numerical computational cost of our method against the brute-force method. Again, we thank the reviewer for this important comment.

---

> > ### Author Response · Authors · 2024-11-27
> >
> > ### **[P2] Numerical Experiment on running time**:
> >
> >
> >
> > We thank the reviewer for this suggestion.
> >
> >
> >
> > The adversarially robust deep learning problem with nonconvex-nonconcave objective enjoys the computational cost reduction and exhibits the trade-off between computation cost v.s. robust solution quality.
> >
> >
> >
> > As shown in `Table 1` of the revised manuscript, for our method, each approximate PPM update, i.e., each Extra gradient update with the full gradient of the training set (ExtraFullGD) generates an approximate robust model in 15 seconds. For the brute-force method, each adversarial training with FGSM (amount the state-of-the-art for fast adversarial training) takes 15 minutes to generate an approximate robust model.
> >
> >
> >
> > The fundamental trade-off: As shown in `Figure 2`, we obtain better-performing approximate robust models when we use better approximates of PPM for our method. This trade-off is mild in adversarial ML. As shown in `Figure 2`, the performance of our method equipped with cheap first-order approximate PPM generates robust models with comparable performance.
> >
> > ---
> >
> > ### **[P3] Generalizability beyond mean-variance optimization**:
> >
> >
> >
> > We thank the reviewer for this comment: the theoretical results are built on robust linear optimization with ellipsoidal uncertainty sets, equivalently, for mean-variance optimization. How does the result generalized?
> >
> >
> >
> > Although our theoretical guarantees are for robust linear optimization with ellipsoidal uncertainty sets, which is equivalent to mean-variance risk measure minimization. This duality result exists in general between uncertainty sets and risk measures [25-27].
> >
> >
> >
> > In this work, we established a particular correspondence between uncertainty sets (ellipsoidal), risk measure (mean-variance) and PPM distance generating function (Mahalanobis), which opens up an important research direction of the co-design between general uncertainty sets and the corresponding PPM distance generating functions.
> >
> >
> >
> > Reference:
> >
> >
> >
> > [25] Natarajan K, Pachamanova D, Sim M. Constructing risk measures from uncertainty sets. Operations research. 2009.
> >
> >
> >
> > [26] Shapiro A, Dentcheva D, Ruszczynski A. Lectures on stochastic programming: modeling and theory. Society for Industrial and Applied Mathematics; 2021.
> >
> >
> >
> > [27] Freund RM. Dual gauge programs, with applications to quadratic programming and the minimum-norm problem. Mathematical Programming. 1987.

---

> > > ### Author Response · Authors · 2024-11-27
> > >
> > > ### **[P4] Closed-form solution to Markowitz Portfolio**:
> > >
> > > Thank you for this suggestion. We included this point in the new manuscript.
> > >
> > > ---
> > >
> > > ### **[P5] Markowitz Portfolio ++, out-of-sample performance**:
> > >
> > > Thank you for this suggestion. We added new discussions on the out-of-sample performance of Markowitz Portfolio ++ in the revised manuscript. The main reason is a significant distributional shift between the in-sample and out-of-sample distributions.
> > >
> > > ---
> > >
> > > ### **[P6] Typos**:
> > >
> > > Thank you for pointing these out. We have made the corrections accordingly.

---

> > > > ### Comment · Reviewer_7fPc · 2024-11-27
> > > >
> > > > Thank you for the detailed clarification. The section on portfolio optimization feels distracting and doesn’t add meaningful insights, so it might be better omitted. While the approximation appears solid, it lacks theoretical backing. It is still not clear why the distribution shift impacts the two out-of-sample performances differently. Therefore, I will maintain the score.

---

> > > > > ### Author Response · Authors · 2024-12-04
> > > > >
> > > > > Thank you for the continued feedback on our work.  We agree the out-of-sample performance discussion is interesting. But it does not add or subtract from our main contributions in this paper. It belongs to another paper.

---

### Official Review · Reviewer_EVSU · 2024-11-04

**Soundness:** 3
**Presentation:** 3
**Contribution:** 2
**Rating:** 5
**Confidence:** 2

**Summary:**

This paper presents a proximal point method based procedure to approximate many Pareto efficient robust solutions. This procedure reduces the computational requirement  by a multiplicity of the number of robust solutions to be generated. They go on to show that this procedure can produce exact Pareto efficient robust solutions for a class of optimization problems.

**Strengths:**

S1 - The paper is generally well written and well presented.

S2 - The technical claims in the paper are generally well explained.

S3 - Experiments are a plus.

S4 - The idea is interesting

**Weaknesses:**

W1 - The literature review is a bit lacking. The comparisons to previous art are inadequate.

W2 - The main novelty is ambiguous.

W3 - The method seems to be limited in its generation of robust solutions where the radius of the uncertainty sets are not freely selectable.

**Questions:**

Q1 - Although the computational efficiency claim somewhat makes sense. I am not sure how this is equivalent to generating multiple robust solutions and comparing their results. While your proximal point method generates pareto efficient intermediate solutions, how many of them does it generate? Why is the new computational complexity $2\times T$ and does not include $N$ at all.

Q2 - How comprehensive is the robust solutions generated by your proximal point based procedure?

Q3 - How to read figure 1 and 2, why are not one-to-one? Are these supposed to be the trajectories followed with the gradient steps?

---

> ### Author Response · Authors · 2024-11-27
>
> We thank the reviewer for the constructive comments. We aim to address all the comments here, please also refer to the revised manuscript with all major updates in blue.
>
> ### **[Answer to W1]: Contribution to the literature**:
>
> We thank the reviewer for raising this point. We agree it is important to improve the positioning of our contribution to the literature. We have added a more extensive literature review in `Section 1` of the revised manuscript. We believe the work is now better positioned within the literature with clear contributions to the state-of-the-art. Specifically, in:
>
> **Continuous Optimization**: The field of continuous optimization predominantly focuses on the question of "how to get to the optimal solution fast" [1-6], not "what does the trajectory as a whole represent". This fundamentally deviates from the existing literature.  Consequently, our work has excited researchers in seminar presentations to the continuous optimization community. Another indirect evidence that we are filling an exciting research gap is that one of the key papers we cited [7] is not highly cited despite its interesting discovery about optimization paths.
>
>
>
> **Implicit Gradient Regularization in ML**: People have explained why and how the iterates of gradient methods, when minimizing a loss function alone, could sometimes provide implicit regularization. We show novel proof and new insights into this problem. Different from the literature [13-20], which tends to be descriptive and focused on unconstrained problems, we present a new direct, constructive proof for the **constrained** setting. This could prove to be an important step forward as constraints naturally appear in many high stakes AI systems: such as safety-constrained reinforcement learning [26-27] for LLM safety alignment [28-30] and autonomous driving [31,32]. Anecdotally, we discovered the connection between our work and this literature after writing our paper. Naturally the perspectives we take and the tools we invent are completely different from the ones already used in the literature, opening up a new way of generalizing results in implicit regularization, including but not limited to constrained settings. Our work demonstrates the implicit regularization can be studied in its dual perspective [21-25] i.e., via robust optimization. Specifically, our work demonstrates approximate proximal point methods (including gradient methods) when minimizing a loss function alone, generates iterates that are approximate robust solutions (as demonstrated in our adversarially robust deep learning experiment). Our result in Corollary 1 also shows that previous unconstrained results can be generalized to heavily constrained problems with polyhedron constraint sets.
>
>
>
> **Robust Optimization**: The authors' have extensive experience publishing and reviewing in robust optimization, beyond our approach, we are not aware of any methods for generating multiple robust solutions other than the naive approach of "solving the problem multiple times" [8-12].
>
>
>
>
>
>
>
> **Action Taken**: We made major revisions to `Section 1` for the revised manuscript, with better positioning of our work within multiple literatures and highlighted our contributions to each literature.
>
>
>
> We want to thank the reviewer again for raising this concern, we believe the work is now well positioned in the literature with clear contributions to the state-of-the-art.
>
>
>
>
>
>
>
> Reference:
>
> Please find the complete list of references included in the global response.

---

> > ### Author Response · Authors · 2024-11-27
> >
> > ### **[Answer to W2 and Q1 Point 1]**
> >
> > We thank the reviewer for bringing up this point: How does our method help overcome the challenge of choosing the radius hyperparameter for robust optimization?
> >
> >  Here we give a clarification of the main novelty and contribution of this work. The ideal practical option for setting the radius hyperparameter in robust optimization is generating multiple robust solutions and comparing their performances. In this procedure, the majority of the computation cost is in generating multiple robust solutions (solving the min-max problem multiple times), comparing their performance comes down to calculating the efficiency (evaluate function value) and the robustness (solve the inner max problem) of the robust solutions which are computationally cheap. Our work provides a new framework to reduce the majority of the cost in setting the radius hyperparameter, i.e., reduce the cost of generating multiple robust solutions.
> >
> > The main novelty of this paper is show for the first time, we can generate multiple (approximate) robust solutions via an (approximate) PPM trajectory. Specifically, we give a new, constructive proof that for constrained robust LPs, exact PPM iterates are exact robust solutions. Subsequently, we give a new algorithm for approximating multiple variable robust solutions as approximate PPM iterates.
> >
> > ---
> >
> > ### **[Answer to Q1 Point 2 and 3]**
> >
> >
> >
> > We thank the reviewer for bringing up these important comments: i) How many robust solutions does Algorithm 1 generate? ii) What is the computational cost of Algorithm 1?
> >
> >
> >
> > **For i)**, in theorem 1, we show under some conditions, every PPM iterates $x_k$ is an exact robust solution with a different radius. Consequently, in algorithm 1, every approximate PPM iterate $x_k$ is an approximate robust solution. Therefore, algorithm 1 can approximate $N$ robust solutions with varying radii by performing $N$ PPM updates.
> >
> >
> >
> > **For ii)** Compared with the cost of the existing method: $N \times T_{\mathrm{RC}}$, the computation cost of our method is $T_{\mathrm{RC}} + (N-1)\times T_{\mathrm{\widetilde{PPM}}}$, where $N$ is the number of robust solutions to be generated,  $T_{\mathrm{RC}}$ is the cost of solving a single robust optimization problem, and $T_{\mathrm{\widetilde{PPM}}}$ is the cost of a single step of an approximate PPM.
> >
> > In general, performing an exact proximal point method update is no easier than solving the robust optimization problem, therefore, the computation cost reduction is enjoyed only when we can equip Algorithm 1 with cheap approximate PPM. Specifically, for linear objective functions, i.e., $f(x,a) = \langle a,x\rangle$, the exact proximal point method updates are equivalent to projected gradient descent (PGD) updates with the same cost as solving the robust problem. For nonlinear differentiable objectives, the proximal point method can be approximated by computationally cheap first-order approximates such as gradient descent, extra-gradient method, and optimistic gradient method [33,38]).
> >
> >
> >
> > Reference:
> >
> >
> >
> > [33] Aryan Mokhtari, Asuman Ozdaglar, and Sarath Pattathil. A unified analysis of extra-gradient and optimistic gradient methods for saddle point problems: Proximal point approach, Proceedings of the Twenty Third International Conference on Artificial Intelligence and Statistics 2020.
> >
> >
> >
> >
> >
> > [38] Parikh N, Boyd S. Proximal algorithms. Foundations and trends® in Optimization. 2014

---

> > > ### Author Response · Authors · 2024-11-27
> > >
> > > ### **[Answer to W3 and Q2]**:
> > >
> > >
> > >
> > > We thank the reviewers for bringing up this important point. The main update is a new closed-form solution to the robust solution radius $\alpha_k(\omega_k)$ for Theorem 1 in `Appendix E` of the revised manuscript. We discuss the potential approach for controlling the granularity of radius $\alpha$.
> > >
> > >
> > >
> > > To show the precise relationship between the PPM iterates and their corresponding robust solution radius. We provide in `Appendix E`, a closed-form solution of $\alpha_{k}$ as $\alpha(\omega_{k},x_{k})= 2\omega_{k} \Vert\Sigma^{1/2}x_{k}\Vert_{2}$, where $\omega_{k}$ is defined by the learning rate sequence, and $x_{k}$ is the current PPM iterate. The practical implication is: given we have computed the current PPM iterate $x_k$, we know $x_k$ is a robust solution with radius $\alpha_k$ which we can calculate in closed-form as a function of $\omega_k$ and $x_k$.
> > >
> > >
> > >
> > >
> > >
> > >
> > >
> > > We are actively working on methods for controlling the robust solution radius sequence $\{\alpha_k\}$ for our next paper. One approach is by controlling the approximate PPM step size, with a smaller step size leading to finer $\alpha_k$ granularity. In practice, we can take initial large approximate PPM steps, and take finer approximate PPM steps once we enter a neighborhood of robust solutions with good efficiency-robustness trade-off for finer robust solution granularity.
> > >
> > >
> > >
> > >
> > >
> > >
> > >
> > > **Action taken**: We give a new closed-formed solution to the robust solution radius $\alpha_k(\omega_k)$ for Theorem 1. We thank the reviewer for this comment as it opens up an important new research direction we hope to build upon the current paper.
> > >
> > > ---
> > >
> > > ### **[Answer to Q3]**.
> > >
> > >
> > >
> > > Thanks for the comment. In Figure 1: The dashed blue lines are trajectories followed by the PPM steps projected onto the nominal return - worst-case return space. The solid red lines are generated by the brute-force approach, i.e., solve the RC multiple times and evaluate nominal return - worst-case return.
> > >
> > >
> > >
> > > Similarly for Figure 2: The dashed lines are trajectories followed by the first-order approximate PPM steps projected onto the clean accuracy - adversarial accuracy space. The solid red lines are generated by the brute-force approach, i.e., solve the adversarial training problem multiple times and evaluate clean accuracy - adversarial accuracy.

---

### Official Review · Reviewer_F2yb · 2024-11-04

**Soundness:** 2
**Presentation:** 2
**Contribution:** 2
**Rating:** 5
**Confidence:** 2

**Summary:**

This paper studies approximating the efficient-robust Pareto solutions through proximal point method. It contributes faster computation requirements compared to the literature.

**Strengths:**

**Originality:**
- Their undertaking of efficient identification of Pareto front seems new.
- The work is a novel combination of well-known techniques.

**Quality:**
- The methods used seem appropriate.
- This is a complete piece of work, with possible future improvements.
- The authors are careful and honest about evaluating both the strengths and weaknesses of their work.

**Clarity:**
- The submission is clearly written.
- It is well organized.

**Significance:**
- The result seems important from a computational efficiency perspective.
- Others (researchers or practitioners) are likely to use the ideas or build on them.
- It provides unique conclusions about existing approaches.

**Weaknesses:**

**Originality:**
- It is not exactly clear how this work differs from previous contributions or if the related work is adequately cited, since there is not substantial comparison.

**Quality:**
- A key point is that the submission has certain issues with technically sound, please see Questions. Similarly, some claims need more support.

**Clarity:**
- It occasionally fails to adequately inform the reader.

**Significance:**
- The difficultness of the task the submission addresses is hard to gauge, considering the rather simple approaches. The comparison with previous work is a bit lacking.
- Thus, it is also not easy to conclude if it advances the state of the art in a demonstrable way.

**Questions:**

**Questions:**
- Line 239: in Theorem 1, $\alpha(\omega_k)$ seems to depend on $x$. How does this work?
- Line 247: what are the two passes exactly?
- Line 299: in Proposition 3, again, the existence of such $\alpha(\omega_k)$ needs to be explained. Is the equality with respect to a specific choice of $x$?

**Major Suggestions:**
- Line 231: define $e$ from the simplex domain.
- Line 329: in Corollary 1, $m$ cannot be arbitrarily large, which makes the probability upper-bounded, so this is not exactly a high probability bound.

**Minor Suggestions:**
- Line 34: need cites in this first paragraph.
- Line 46: also need more cites in the second paragraph.
- Line 91: $E$ is not empty but only includes the zero vector.
- Line 173: explain the connection of this central path to the robust optimization problem earlier on.

---

> ### Author Response · Authors · 2024-11-27
>
> We thank the reviewer for the constructive comments. We aim to address each comment here, please also refer to the revised manuscript with major updates in blue.
> ### **[P1]: Contribution to the literature**:
>
> We thank the reviewer for raising this point. We agree it is important to improve the positioning of our contribution to the literature. We have added a more extensive literature review in `Section 1` of the revised manuscript. We believe the work is now better positioned within the literature with clear contributions to the state-of-the-art. Specifically, in:
>
>
>
>
>
> **Continuous Optimization**: The field of continuous optimization predominantly focuses on the question of "how to get to the optimal solution fast" [1-6], not "what does the trajectory as a whole represent". This fundamentally deviates from the existing literature.  Consequently, our work has excited researchers in seminar presentations to the continuous optimization community. Another indirect evidence that we are filling an exciting research gap is that one of the key papers we cited [7] is not highly cited despite its interesting discovery about optimization paths.
>
>
>
> **Implicit Gradient Regularization in ML**: People have explained why and how the iterates of gradient methods, when minimizing a loss function alone, could sometimes provide implicit regularization. We show novel proof and new insights into this problem. Different from the literature [13-20], which tends to be descriptive and focused on unconstrained problems, we present a new direct, constructive proof for the **constrained** setting. This could prove to be an important step forward as constraints naturally appear in many high stakes AI systems: such as safety-constrained reinforcement learning [26-27] for LLM safety alignment [28-30] and autonomous driving [31,32]. Anecdotally, we discovered the connection between our work and this literature after writing our paper. Naturally the perspectives we take and the tools we invent are completely different from the ones already used in the literature, opening up a new way of generalizing results in implicit regularization, including but not limited to constrained settings. Our work demonstrates the implicit regularization can be studied in its dual perspective [21-25] i.e., via robust optimization. Specifically, our work demonstrates approximate proximal point methods (including gradient methods) when minimizing a loss function alone, generates iterates that are approximate robust solutions (as demonstrated in our adversarially robust deep learning experiment). Our result in Corollary 1 also shows that previous unconstrained results can be generalized to heavily constrained problems with polyhedron constraint sets.
>
>
>
> **Robust Optimization**: The authors' have extensive experience publishing and reviewing in robust optimization, beyond our approach, we are not aware of any methods for generating multiple robust solutions other than the naive approach of "solving the problem multiple times" [8-12].
>
>
>
>
>
>
>
> **Action Taken**: We made major revision to `Section 1` for the revised manuscript, with better positioning of our work within multiple literatures and highlighted our contributions to each literature.
>
>
>
> We want thank the reviewer again for raising this concern, we believe the work is now well positioned in the literature with clear contribution to the state-of-the-art.
>
>
>
>
>
>
>
> Reference:
>
>
>
> Please find the complete list of references included in the global response.

---

> ### Author Response · Authors · 2024-11-27
>
> ### **[P2] Theorem 1 ($\alpha(\omega_k)$)**:
>
> We thank the reviewer for raising this important point. We have added a new closed-form solution to the robust solution radius $\alpha_k$ in `Appendix E` of the revised manuscript, a new closed-form solution of $\alpha_{k}$ as $\alpha(\omega_{k},x_{k})= 2\omega_{k} \Vert\Sigma^{1/2}x_{k}\Vert_{2}$, where $\omega_{k}$ is defined by the learning rate sequence, and $x_{k}$ is the current PPM iterate. The practical implication is: given we are currently at PPM iterate $x_k$, we know $x_k$ is a robust solution with radius $\alpha_k$ which we can calculate in closed-form.
>
> ---
>
> ### **[P3] Two algorithmic passes in Algorithm 1**:
>
>
>
> We thank the reviewer for this comment.
>
>
>
> **First algorithmic pass**: Solve for the robust solution $x_{\mathrm{R}}=\arg \min_{x\in\mathcal{X}}\max_{a \in \mathcal{U}} \ f(x,a)$. In practice, the shape of $\mathcal{U}$ is designed according to the specific problem (e.g. ellipsoidal for uncertain portfolio returns, box for adversarial image attack), its radius can be set sufficiently large in order for algorithm 1 to cover a large range of $r$. To compute such $x_R$, tractable exact methods exist [22,23] for convex-concave $f$, and tractable approximate methods exist [25,26] for nonconvex-nonconcave $f$, such as in our adversarially robust deep learning experiment.
>
>
>
>
>
>
>
> **Second algorithmic pass**: Solve the nominal problem $min_{x\in\mathcal{X}}f(x,a_0)$ with approximate PPM, initialized by the robust solution, $x_{\mathrm{R}}$. Specifically, set $x_0=x_{\mathrm{R}}$, iteratively perform approximate PPM update: $x_{k+1} \approx \arg\min_{x\in \mathcal{X}} f(x, a_{0}) +\lambda_{k} D_{\varphi}(x,x_{k})$.
>
>
>
> Finally, the (approximate) PPM sequence $\{ x_k \}$ are (approximate) robust solutions $\{x_{\mathrm{PE}}(\alpha_k)\}$, where $\alpha_k = 2\omega_{k} \Vert\Sigma^{1/2}x_{k}\Vert_{2}$ for all $k$.
>
>
>
>
>
>
>
> Reference:
>
>
>
> [22] Aharon Ben-Tal, Laurent El Ghaoui, and Arkadi Nemirovski. Robust Optimization. Princeton University Press, 1 edition, 2009.
>
>
>
> [23] A. Ben-Tal and A. Nemirovski. Robust solutions of uncertain linear programs. Operations Research Letters, 25:1–13, 8 1999. ISSN 01676377. doi: 10.1016/S0167-6377(99)00016-4.
>
>
>
> [25] Aleksander Madry, Aleksandar Makelov, Ludwig Schmidt, Dimitris Tsipras, and Adrian Vladu. Towards
>
>
>
> deep learning models resistant to adversarial attacks. In ICLR, 2018.
>
>
>
>
>
> [26] Eric Wong, Leslie Rice, and J. Zico Kolter. Fast is better than free: Revisiting adversarial training. In ICLR, 2020.
>
> ---
>
> ### **[P4] Notations**
>
> Thanks for the comment. $e$ represents a vector of ones. We include the definition of all notations in section `2.1. Notations`.
>
> ---

---

> > ### Author Response · Authors · 2024-11-27
> >
> > ### **[P5] Probabilistic bound**
> >
> > We thank the reviewer for bringing up this important point. Different from the implicit gradient regularization literature that studies unconstraint problems [13-20], the main contribution of our work is a novel, constructive proof for constrained problems. Therefore, we positioned the main application of Corollary 1 towards heavily constrained robust optimization problems with large $m$ (e.g. robust SVM [22], risk-measure minimization [25], robust MDP [26,27] and mechanism design [28]).
> >
> > Reference:
> >
> > [13] Barrett, D. and Dherin, B. Implicit gradient regularization. ICLR, 2021.
> >
> > [14] Suriya Gunasekar, Jason Lee, Daniel Soudry, and Nathan Srebro. Characterizing implicit bias in terms of optimization geometry. ICML, pages 1832–1841. PMLR, 2018.
> >
> > [15] Haoyuan Sun, Khashayar Gatmiry, Kwangjun Ahn, Navid Azizan, A Unified Approach to Controlling Implicit Regularization via Mirror Descent ICML, 2023.
> >
> > [16] Ziwei Ji and Matus Telgarsky. The implicit bias of gradient descent on nonseparable data. In Conference on Learning Theory, pages 1772–1798. PMLR, 2019a.
> >
> > [17] Yan Li, Caleb Ju, Ethan X Fang, and Tuo Zhao. Implicit regularization of bregman proximal point algorithm and mirror descent on separable data. arXiv preprint arXiv:2108.06808, 2021.
> >
> > [18] Ziwei Ji, Miroslav Dud.k, Robert E. Schapire, and Matus Telgarsky. Gradient descent follows the regularization path for general losses. In Proceedings of Thirty Third Conference on Learning Theory (Proceedings of Machine Learning Research, Vol. 125) 2020
> >
> > [19] Arun Suggala, Adarsh Prasad, and Pradeep K Ravikumar, Connecting Optimization and Regularization Paths. In NIPS, 2018.
> >
> > [20]  Jingfeng Wu, Vladimir Braverman, and Lin Yang, Obtaining Adjustable Regularization for Free via Iterate Averaging. In ICML 2020.
> >
> > [22] Xu H, Caramanis C, Mannor S. Robustness and Regularization of Support Vector Machines. Journal of machine learning research. 2009.
> >
> >
> >
> > [25] Natarajan K, Pachamanova D, Sim M. Constructing risk measures from uncertainty sets. Operations research. 2009.
> >
> > [26] Iyengar GN. Robust dynamic programming. Mathematics of Operations Research. 2005
> >
> > [27] El Housni O, Goyal V. Beyond worst-case: A probabilistic analysis of affine policies in dynamic optimization. NIPS. 2017
> >
> > [28] Vohra RV. Mechanism design: a linear programming approach. Cambridge University Press; 2011
> >
> > ---
> >
> >
> >
> > ### **[P6] Minor Suggestions**
> >
> > We thank the reviewer for the suggestions. We have referenced 27 additional papers for the extended literature review in `Section 1` of the revised manuscript, and improved the positioning of our work in the literature, as well as discussing the contributions of our work to the literature. We have corrected the discussion for $\Xi$, $\Xi$ is a singleton for the nominal problem. We have added additional discussion on the relationship between the central path and the robust solutions in `line 215` of the revised manuscript.

---

> ### Comment · Reviewer_F2yb · 2024-12-03
>
> Thank you for the rebuttal.
>
> Regarding $m$ in Corollary 1, it is inherently bounded due to $\epsilon<1$. Hence, the probability cannot be pushed to $1$. Furthermore, large $m$ results in large $\epsilon$ and, thus, large range of performance values. This weakens the approximation.
>
> I have gone through all reviews and responses. An issue seemingly arises in determining how meaningfully your main findings, which are tailored to the linear case (which itself does not enjoy computational benefits), should be interpreted for the nonlinear case, where you claim computational benefits.

---

> > ### Author Response · Authors · 2024-12-04
> >
> > Thank you for taking the time to continue to engage with our work and providing constructive feedback.
> >
> > [P1] For $m$ sufficiently large and for $n$ sufficiently larger than $\log(m)$, the approximation is tight. Note that $m \geq n \geq \log(m)$ is a reasonable condition in constrained optimization problems.
> >
> > [P2] Thank you for raising this important point. The meaning of the main finding lies in the structural understanding of what the set of Pareto robust solutions look like – our results give a constructive approach (via PPM) to produce / approximate such trajectory under different cases. These cases are more general than classical results that appear in multiple literatures (two-fund theorem in finance, regularization paths in stats), in the sense that our results deal with a constrained setting instead of unconstrained settings. For whether the constructive PPM path enjoys computational benefits, we have given a more precise statement in the revision to reflect the review team’s comments, and will leave the detailed analysis of approximation-tractability tradeoffs to future papers since it is another nuanced topic, with many combinations of problem instances and PPM approximations available on the market.

---

### Official Review · Reviewer_qiW4 · 2024-11-07

**Soundness:** 2
**Presentation:** 2
**Contribution:** 2
**Rating:** 5
**Confidence:** 2

**Summary:**

This paper studies robust optimization with an uncertainty set, and proposes a more efficient way for computing Pareto efficient robust solutions based on proximal point methods.

**Strengths:**

This paper proposes a novel way for solving robust optimization with an uncertainty set. The idea of applying Proximal point method for computing the Pareto efficient robust solutions seems to be a novel idea that has been explored before.

Empirical results demonstrate the effectiveness of the proposed methods.

**Weaknesses:**

A limited setting: The algorithms and theoretical guarantees given by the authors seem to only work on a toy example with linear functions and ellipsoidal uncertainty set. The objective function (e.g., in (3)) has a very particular form, and it seems that the analysis is difficult to be generalized to other cases.

One of the main advantage mentioned by the authors is that the total computational cost is reduced from NT to 2T. I wonder can the authors provide more discussion on this point? To be more specific, which papers are we talking here (Line 054)?

I am not sure how to understand Theorem 1. The algorithm gives a series of x_k, which are PE in terms of  a series of special alpha (i.e., alpha(\omega_k)). What are these alpha(\omega_k)? how do we know this sequence is good enough? That is, how do we know these alpha covers enough possible alphas, and how does alpha(\omega_k) varies with \omega_k?

Line 239:  (alpha(\omega_k)) is such that the equality holds. How do we know such an alpha exist? How do we compute it?

I have some doubts on how to implement the proposed methods. Specifically: Line 234: when computing x_R, \xi\in\Xi(\infty). So how exactly should we compute x_R? What is \mathcal{U} in Line 253 for the linear problem with ellipsoidal uncertainty set?

**Questions:**

See above.

---

> ### Author Response · Authors · 2024-11-27
>
> We are glad that the reviewer finds our approach novel, and we appreciate the comment from the reviewer that led us to improve the paper in the following aspects.
>
> - Additional discussion on the computation cost of the algorithm, in `section 4.2`.
>
> - A new closed-form solution to the robust solution radius $\alpha_k$, in `Appendix E`.
>
> We will now address each comment in detail.
>
> ### **[P1] Generalizability**
>
> We thank the reviewers for bringing up these concerns: i) How generalizable is the theoretical guarantee for robust linear optimization with constraints? ii) How generalizable is the subsequent PPM-based algorithm for approximating multiple robust solutions?
>
> The problem of robust optimization with linear objective function (in x), polyhedral constraints (in x), and with general convex uncertainty sets (in u) is the central object of study in Robust Optimization [8,9].  Many problems can be formulated exactly as (robust) linear optimization with constraints (e.g., risk-measure minimization [34], robust MDP [35,36] and mechanism design [37]). Linear optimization with constraints poses additional theoretical complexity compared with unconstrained convex optimization problems, due to the combinatorial nature of its polyhedron constraint set (e.g., the simplex algorithm exploits the combinatorial nature of LPs, it is empirically among the most efficient algorithms for solving LPs despite its exponential worst-case complexity, it has been a challenging question explaining its empirical performance).
>
> Our theoretical results (Corollary 1 and Theorem 1) consider robust linear optimization problems with general polyhedron feasible region and ellipsoidal uncertainty set. Building upon the theoretical results, our result generalizes to general convex uncertainty sets and more general objective functions. Specifically,
>
> - Our analysis reveals a new insight: there is a correspondence between the shape of the uncertainty set (ellipsoidal uncertainty set) and the distance-generating function in PPM (Mahalanobis distance), this correspondence exists in general and poses a new research direction, and we are actively investigating the co-design of uncertainty set and the distance-generating function building upon this paper.
>
> - In the adversarially robust deep learning experiment, we show empirically, our method generalized to nonconvex-nonconcave objective functions (our method is 60 times faster than the brute-force method in generating each robust model `[Table 1]`, with comparable model performance `[Figure 2]`).
>
>
>
> Reference:
>
> [8] Aharon Ben-Tal, Laurent El Ghaoui, and Arkadi Nemirovski. Robust Optimization. Princeton University Press, 1 edition, 2009.
>
> [9] A. Ben-Tal and A. Nemirovski. Robust solutions of uncertain linear programs. Operations Research Letters, 25:1–13, 8 1999. ISSN 01676377. doi: 10.1016/S0167-6377(99)00016-4.
>
> [34] Natarajan K, Pachamanova D, Sim M. Constructing risk measures from uncertainty sets. Operations research. 2009.
>
> [35] Iyengar GN. Robust dynamic programming. Mathematics of Operations Research. 2005
>
> [36] El Housni O, Goyal V. Beyond worst-case: A probabilistic analysis of affine policies in dynamic optimization. NIPS. 2017
>
> [37] Vohra RV. Mechanism design: a linear programming approach. Cambridge University Press; 2011

---

> > ### Author Response · Authors · 2024-11-27
> >
> > ### **[P2] Computation Cost**:
> >
> > We thank the reviewer for bringing up this point. To the best of our knowledge, there are no methods in the literature for generating multiple robust solutions other than “solving the problem multiple times” [8-12]. Our method generates multiple approximate robust solutions via approximate PPM iterates.
> >
> >
> >
> > More precisely, the computation cost of algorithm 1 is $ T_{\mathrm{R}} + (N-1)T_{\mathrm{\widetilde{PPM}}}$, where $T_{\mathrm{R}}$ is the cost of solving a single robust optimization problem for initializing the PPM, $T_{\mathrm{\widetilde{PPM}}}$ is the cost of a single approximate PPM update (cheap first-order methods such as GD, Extra-Gradient can be used in practice as approximate PPM [33,38]) generating one approximate robust solution per approximate PPM update. Consequently, we need to trade off between computation cost v.s. robust solution quality, in shorter, better approximation to PPM update generates better approximate robust solutions but at a higher cost. We observe such trade-off in our adversarially robust deep learning experiments. As shown in `Figure 2`, we obtain better-performing approximate robust models when we use better approximates of PPM for our method. In addition, we show this trade-off is mild in practice, our method with the best (highest cost) approximate PPM (full gradient descent) is **60 times faster** than traditional adversarial training for generating robust models **without sacrificing too much model performance**. As shown in `Table 1`, our method with Extra-gradient method as approximate PPM takes **15 seconds** to generate one adversarial robust model v.s. traditional adversarial robust training that takes **15 minutes** per adversarial robust model. As shown in `Figure 2`, the performance of our method against traditional adversarial robust training is comparable.
> >
> >
> >
> > **Action Taken**: we give a more detailed discussion on the computation cost of our approach in `Sections 4.2.` of the revised manuscript.
> >
> >
> > Reference
> >
> > [8] Aharon Ben-Tal, Laurent El Ghaoui, and Arkadi Nemirovski. Robust Optimization. Princeton University Press, 1 edition, 2009.
> >
> >
> >
> > [9] A. Ben-Tal and A. Nemirovski. Robust solutions of uncertain linear programs. Operations Research Letters, 25:1–13, 8 1999. ISSN 01676377. doi: 10.1016/S0167-6377(99)00016-4.
> >
> >
> >
> > [10] Aharon Ben-Tal, Stephen Boyd, and Arkadi Nemirovski. Extending scope of robust optimization: Comprehensive robust counterparts of uncertain problems. Mathematical Programming, 107:63–89, 6 2006. ISSN 0025-5610. doi: 10.1007/s10107-005-0679-z.
> >
> >
> >
> > [11] Dan A. Iancu and Nikolaos Trichakis. Pareto efficiency in robust optimization. Management Science, 60: 130–147, 1 2014. ISSN 0025-1909. doi: 10.1287/mnsc.2013.1753.
> >
> >
> >
> > [12] Dimitris Bertsimas and Melvyn Sim. The price of robustness. Operations Research, 52:35–53, 2 2004. ISSN 0030-364X. doi: 10.1287/opre.1030.0065.
> >
> >
> >
> > [33] Aryan Mokhtari, Asuman Ozdaglar, and Sarath Pattathil. A unified analysis of extra-gradient and optimistic gradient methods for saddle point problems: Proximal point approach, Proceedings of the Twenty Third International Conference on Artificial Intelligence and Statistics 2020.
> >
> >
> >
> > [38] Parikh N, Boyd S. Proximal algorithms. Foundations and trends® in Optimization. 2014

---

> ### Author Response · Authors · 2024-11-27
>
> ### **[P3] $\{\alpha(\omega_k)\}$ sequence**:
>
> We thank the reviewer for bringing up this important point. We agree it is important to know exactly the $\{\alpha(\omega_k)\}$ sequence generated from the PPM sequence. Towards this, we provide in `Appendix E` of the revised manuscript, a new closed-form solution of $\alpha_{k}$ as $\alpha(\omega_{k},x_{k})= 2\omega_{k} \Vert\Sigma^{1/2}x_{k}\Vert_{2}$, where $\omega_{k}$ is defined by the learning rate sequence, and $x_{k}$ is the current PPM iterate. The practical implication is: given we are currently at PPM iterate $x_k$, we know $x_k$ is a robust solution with radius $\alpha_k$ which we can calculate in closed-form.
>
> ---
>
> ### **[P4] Implementing our algorithm**:
>
> **P4.1.) Compute $x_{\mathrm{R}}$**: In short, from the literature, we have tractable exact solutions for $x_{\mathrm{R}}$ for a large class of problems, and tractable approximate solutions for nonconvex-nonconcave objectives. Solving (RC) exactly comes down to finding its computationally tractable reformulation which is typically its Fenchel dual [21]. Such tractable reformulation exists for robust LP [22,23] and more generally for robust nonlinear optimization problems [22]. In algorithm 1, depending on the range of radius, $r$ we want to cover, we can start with a $x_{\mathrm{R}}$ induced by $\mathcal{U} (\infty)$ or a sufficiently large $\mathcal{U} (r_\mathrm{max})$ with a large $r_{\mathrm{max}}$, setting  $r$ to $\infty$ is equivalent to set the regularization term weight to $\infty$ in the dual problem (e.g. line 247 of the revised manuscript, reducing mean-standard deviation risk measure minimization to just standard deviation minimization), hence an exact solution to $x_{\mathrm{R}}$ with remains tractable for $\mathcal{U} (\infty)$. For nonconvex-nonconcave objective functions such as in our adversarially robust deep learning example, approximation algorithms exist [25,26] for approximating the initial $x_{\mathrm{R}}$.
>
> **P4.2.) Uncertainty set design**: Although our theoretical results consider ellipsoidal uncertainty sets, algorithm 1 applies to general uncertainty sets (e.g. $L\infty$ norm ball uncertainty set in our adversarially robust deep learning experiment). Theorem 1 shows there is a correspondence between uncertainty set, $\mathcal{U}$ and the Bregman distance in PPM (we show ellipsoidal $\mathcal{U}$ correspond Bregman distance induced by the Mahalanobis distance). This correspondence is more general and opens up interesting new research that we are actively investigating for our next paper: designing the PPM Bregman distance according to specific $\mathcal{U}$.
>
>  Reference
>
> [21]  Rockafellar RT. Convex analysis. Princeton NJ, USA: Princeton University Press; 1997.
>
> [22] Aharon Ben-Tal, Laurent El Ghaoui, and Arkadi Nemirovski. Robust Optimization. Princeton University Press, 1 edition, 2009.
>
> [23] A. Ben-Tal and A. Nemirovski. Robust solutions of uncertain linear programs. Operations Research Letters, 25:1–13, 8 1999. ISSN 01676377. doi: 10.1016/S0167-6377(99)00016-4.
>
> [24] Ben-Tal, A., den Hertog, D. & Vial, JP. Deriving robust counterparts of nonlinear uncertain inequalities.Math. Program. 149, 265–299 (2015)
>
> [25] Aleksander Madry, Aleksandar Makelov, Ludwig Schmidt, Dimitris Tsipras, and Adrian Vladu. Towards
>
> deep learning models resistant to adversarial attacks. In ICLR, 2018.
>
>
>
> [26] Eric Wong, Leslie Rice, and J. Zico Kolter. Fast is better than free: Revisiting adversarial training. In
>
> ICLR, 2020.

---

### Author Response · Authors · 2024-11-27
**Global Response**

Dear Reviewers:

We would like to thank the reviewers for their constructive comments on our work. We have addressed all the comments in the `revised manuscript` (with major updates highlighted in blue), as a result, we believe it is now significantly improved.

Before addressing each reviewer’s comments in detail. For your convenience, we summarized below a global response to the main comments and the major updates we made to address them.

## **[Q1]: Position in the literature**

We thank the reviewers for raising these concerns: i) How is this work positioned within the literature beyond robust optimization? ii) What is the unique contribution of this work to these different literatures? We address these questions positively below.

**A1: Additional literature review and highlighting our contribution**: Given the nature of the problem, this work is positioned at the intersection of three main literatures, our work contributes to each literature in a substantial and novel way, opening up meaningful future research directions.

**Continuous Optimization**: The field of continuous optimization predominantly focuses on the question of "how to get to the optimal solution fast" [1-6], not "what does the trajectory as a whole represent". This fundamentally deviates from the existing literature.  Consequently, our work has excited researchers in seminar presentations to the continuous optimization community. Another indirect evidence that we are filling an exciting research gap is that one of the key papers we cited [7] is not highly cited despite its interesting discovery about optimization paths.

**Implicit Gradient Regularization in ML**: People have explained why and how the iterates of gradient methods, when minimizing a loss function alone, could sometimes provide implicit regularization. We show novel proof and new insights into this problem. Different from the literature [13-20], which tends to be descriptive and focused on unconstrained problems, we present a new direct, constructive proof for the **constrained** setting. This could prove to be an important step forward as constraints naturally appear in many high-stakes AI systems: such as safety-constrained reinforcement learning [26-27] for LLM safety alignment [28-30] and autonomous driving [31,32]. Anecdotally, we discovered the connection between our work and this literature after writing our paper. Naturally, the perspectives we take and the tools we invent are completely different from the ones already used in the literature, opening up a new way of generalizing results in implicit regularization, including but not limited to constrained settings. Our work demonstrates that implicit regularization can be studied in its dual perspective [21-25] i.e., via robust optimization. Specifically, our work demonstrates approximate proximal point methods (including gradient methods) when minimizing a loss function alone, generates iterates that are approximate robust solutions (as demonstrated in our adversarially robust deep learning experiment). Our result in Corollary 1 also shows that previous unconstrained results can be generalized to heavily constrained problems with polyhedron constraint sets.

**Robust Optimization**: The authors have extensive experience publishing and reviewing in robust optimization, beyond our approach, we are not aware of any methods for generating multiple robust solutions other than the naive approach of "solving the problem multiple times" [8-12].



**Action Taken**: We made major revisions to `Section 1`, with better positioning of our work within multiple literatures, and highlighted our contributions to each literature.

We want to thank the reviewer again for raising this concern, we believe the work is now well positioned in the literature with clear contributions to the state-of-the-art.

---

> ### Author Response · Authors · 2024-11-27
> **Reference**
>
> Reference:
>
> [1] Corman, E., & Yuan, X.. A generalized proximal point algorithm and its convergence rate. SIAM J. Optim., 2014.
>
> [2]  O. Güler. On the convergence of the proximal point algorithm for convex minimization. SIAM J. Control Optim., 1991.
>
> [3]  O. Güler. New proximal point algorithms for convex minimization. SIAM J. Optim., 1992.
>
> [4]  R.T. Rockafellar. Augmented Lagrangians and applications of the proximal point algorithm in convex programming. Math. Oper. Res., , 1976.
>
> [5]  R.T. Rockafellar. Monotone operators and the proximal point algorithm. SIAM J. Control Optim., 1976.
>
> [6] H. Lu, R. Freund, and Y. Nesterov, Relatively smooth convex optimization by first-order methods, and applications, SIAM J. Optim., 28 (2018)
>
> [7] Alfredo N. Iusem, B. F. Svaiter, and João Xavier da Cruz Neto, Central Paths, Generalized Proximal Point Methods, and Cauchy Trajectories in Riemannian Manifolds, SIAM J. Control Optim. 1999
>
> [8] Aharon Ben-Tal, Laurent El Ghaoui, and Arkadi Nemirovski. Robust Optimization. Princeton University Press, 1 edition, 2009.
>
> [9] A. Ben-Tal and A. Nemirovski. Robust solutions of uncertain linear programs. Operations Research Letters 1999
>
> [10] Aharon Ben-Tal, Stephen Boyd, and Arkadi Nemirovski. Extending the scope of robust optimization: Comprehensive robust counterparts of uncertain problems. Mathematical Programming, 2006
>
> [11] Dan A. Iancu and Nikolaos Trichakis. Pareto efficiency in robust optimization. Management Science 2014.
>
> [12] Bertsimas D. and Sim M. The price of robustness. Operations Research 2004.
>
> [13] Barrett, D. and Dherin, B. Implicit gradient regularization. ICLR, 2021.
>
> [14] Gunasekar S, Lee J, Soudry D, and Srebro N. Characterizing implicit bias in terms of optimization geometry. ICML 2018.
>
> [15] Haoyuan Sun, Khashayar Gatmiry, Kwangjun Ahn, Navid Azizan, A Unified Approach to Controlling Implicit Regularization via Mirror Descent ICML, 2023.
>
> [16] Ziwei Ji and Matus Telgarsky. The implicit bias of gradient descent on nonseparable data. In Conference on Learning Theory, pages 1772–1798. PMLR, 2019.
>
> [17] Yan Li, Caleb Ju, Ethan X Fang, and Tuo Zhao. Implicit regularization of bregman proximal point algorithm and mirror descent on separable data. arXiv preprint 2021.
>
> [18] Ziwei Ji, Miroslav Dud.k, Robert E. Schapire, and Matus Telgarsky. Gradient descent follows the regularization path for general losses. In Proceedings of Thirty Third Conference on Learning Theory 2020
>
> [19] Arun Suggala, Adarsh Prasad, and Pradeep K Ravikumar, Connecting Optimization and Regularization Paths. In NIPS, 2018.
>
> [20]  Jingfeng Wu, Vladimir Braverman, and Lin Yang, Obtaining Adjustable Regularization for Free via Iterate Averaging. In ICML 2020.
>
> [21] El Ghaoui L, Lebret H. Robust solutions to least-squares problems with uncertain data. SIAM Journal on matrix analysis and applications. 1997.
>
> [22] Xu H, Caramanis C, Mannor S. Robustness and Regularization of Support Vector Machines. Journal of machine learning research. 2009.
>
> [23] Shapiro A, Dentcheva D, Ruszczynski A. Lectures on stochastic programming: modeling and theory.
>
> [24] Freund RM. Dual gauge programs, with applications to quadratic programming and the minimum-norm problem. Mathematical Programming. 1987.
>
> [25] Natarajan K, Pachamanova D, Sim M. Constructing risk measures from uncertainty sets. Operations research. 2009.
>
> [26] Achiam J, Held D, Tamar A, Abbeel P. Constrained policy optimization. ICML 2017
>
> [27] Yang Q, Simão TD, Tindemans SH, Spaan MT. WCSAC: Worst-case soft actor-critic for safety-constrained reinforcement learning. AAAI 2021.
>
> [28] Wachi A, Tran TQ, Sato R, Tanabe T, Akimoto Y. Stepwise alignment for constrained language model policy optimization. arXiv preprint 2024
>
> [29] Liu Z, Sun X, Zheng Z. Enhancing llm safety via constrained direct preference optimization. arXiv preprint 2024
>
> [30] Dai J, Pan X, Sun R, Ji J, Xu X, Liu M, Wang Y, Yang Y. Safe RLHF: Safe reinforcement learning from human feedback. arXiv preprint 2023
>
> [31] Wen L, Duan J, Li SE, Xu S, Peng H. Safe reinforcement learning for autonomous vehicles through parallel constrained policy optimization. In ITSC 2020
>
> [32] Gu S, Yang L, Du Y, Chen G, Walter F, Wang J, Knoll A. A Review of Safe Reinforcement Learning: Methods, Theories and Applications. IEEE Transactions on Pattern Analysis and Machine Intelligence. 2024
>
> [33] Mokhtari A, O Asuman, and P Sarath. A unified analysis of extra-gradient and optimistic gradient methods for saddle point problems: Proximal point approach, AISTATS 2020.
>
> [34] Natarajan K, Pachamanova D, Sim M. Constructing risk measures from uncertainty sets. Operations research. 2009.
>
> [35] Iyengar GN. Robust dynamic programming. Mathematics of Operations Research. 2005
>
> [36] El Housni O, Goyal V. Beyond worst-case: A probabilistic analysis of affine policies in dynamic optimization. NIPS. 2017
>
> [37] Vohra RV. Mechanism design: a linear programming approach. 2011

---

### Author Response · Authors · 2024-11-27
**Global Response (Cont'd)**

## **[Q2]: Theoretical Guarantee is on linear objective, simplex constraint set and ellipsoidal uncertainty set. Questions for generalizability.**

**A2: Generalizability**: We thank the reviewers for bringing up these concerns: i) How generalizable is the theoretical guarantee for robust linear optimization with constraints? ii) How generalizable is the subsequent PPM-based algorithm for approximating multiple robust solutions?

The problem of robust optimization with linear objective function (in x), polyhedral constraints (in x), and with general convex uncertainty sets (in u) is the central object of study in Robust Optimization [8,9].  Many problems can be formulated exactly as (robust) linear optimization with constraints (e.g., risk-measure minimization [34], robust MDP [35,36] and mechanism design [37]). Linear optimization with constraints poses additional theoretical complexity compared with unconstrained convex optimization problems, due to the combinatorial nature of its polyhedron constraint set (e.g., the simplex algorithm exploits the combinatorial nature of LPs, it is empirically among the most efficient algorithms for solving LPs despite its exponential worst-case complexity, it has been a challenging question explaining its empirical performance).

Our theoretical results (Corollary 1 and Theorem 1) consider robust linear optimization problems with general polyhedron feasible region and ellipsoidal uncertainty set. Building upon the theoretical results, our result generalizes to general convex uncertainty sets and more general objective functions. Specifically,

-Our analysis reveals a new insight: there is a correspondence between the shape of the uncertainty set (ellipsoidal uncertainty set) and the distance-generating function in PPM (Mahalanobis distance), this correspondence exists in general and poses a new research direction, and we are actively investigating the co-design of uncertainty set and the distance-generating function building upon this paper.

-In the adversarially robust deep learning experiment, we show empirically, our method generalized to nonconvex-nonconcave objective functions (our method is 60 times faster than the brute-force method in generating each robust model `[Table 1]`, with comparable model performance `[Figure 2]`).

---

### Author Response · Authors · 2024-11-27
**Global Response (Cont'd)**

## **[Q3]: Computation cost**

**A3**: We thank the reviewers for raising this important point. We provide a more detailed discussion of the computation cost of our method v.s. the brute-force method.

First, as discussed in line 260 of the original manuscript, and as correctly pointed out by the review team, in general, an exact PPM update is no easier to solve than the corresponding robust optimization solution. Hence, in practice, we need to compute cheap approximate PPM updates (e.g., gradient methods) that generate approximate robust solutions, trading-off between computation cost v.s. robust solution quality. The major updates expanding on this point are:

The more precise computation cost of algorithm 1 is: $ T_\mathrm{R} + (N-1)T_{\mathrm{\widetilde{PPM}}}$, where $T_\mathrm{R}$ is the cost of solving the robust optimization problem, $T_{\mathrm{\widetilde{PPM}}}$ is the cost of a single (approximate) PPM update.



The computational cost reduction can be enjoyed under the condition that objective function is nonlinear and (sub)differentiable, for which we can use first-order approximate PPM that is cheaper than solving the robust optimization problem.



The fundamental trade-off in our approach is between computation cost v.s. robust solution quality. Our approach provides a practical lever to adjust this trade-off by adjusting the choice of the approximate PPM. In short, better approximate PPM leads to better robust solution quality but at a higher cost. Cheap first-order methods such as gradient descent, mirror descent and extra-gradient Method can all be considered as approximate PPM with only the first-order information [21].



Such a trade-off is observed empirically in our adversarially robust deep learning experiments. As shown in Figure 2, we obtain better-performing approximate robust models when we use better approximates of PPM for our method. In addition, we show this trade-off is mild in adversarially robust deep learning, our method with the best (highest cost) approximate PPM (full gradient descent) is **60 times faster** than traditional adversarial training for generating robust models **without sacrificing too much model performance**. As shown in Table 1, our method with the Extra-gradient method as approximate PPM takes **15 seconds** to generate one adversarial robust model v.s. traditional adversarial robust training that takes **15 mins** per adversarial robust model. As shown in Figure 2, the performances of our method against traditional adversarial robust training are comparable.



**Action Taken**: Thanks to the reviewers’ comments, we made major revisions to the computation cost of our approach throughout `Sections 1 and 4`, introducing more precisely the computation cost of algorithm 1 and highlighting the fundamental trade-off between the computation cost v.s. robust solution quality and the lever for adjusting this trade-off via the choice of the approximate PPM.

---

### Author Response · Authors · 2024-11-27
**Global Response (Cont'd)**

## **[Q4]: How to control the granularity of the radius of PPM-generated robust solutions? How comprehensive is the robust solutions generated by the PPM procedure?**



**A4**: We thank the reviewers for bringing up these questions. We agree these are important points for the deployment of our method. The main update is a new closed-form solution to the robust solution radius $\alpha_k(\omega_k)$ for Theorem 1. We discuss the potential approach for controlling the granularity of radius $alpha$.



To show the precise relationship between the PPM iterates and their corresponding robust solution radius. We provide in `Appendix E`, a closed-form solution of $\alpha_{k}$ as $\alpha(\omega_{k},x_{k}) = 2 \omega_{k} \Vert \Sigma^{1/2} x_{k} \Vert_{2}$, where $\omega_{k}$ is defined by the learning rate sequence, and $x_{k}$ is the current PPM iterate. The practical implication is: given we have computed the current PPM iterate $x_k$, we know $x_k$ is a robust solution with radius $\alpha_k$ which we can calculate in closed-form as a function of $\omega_k$ and $x_k$.



We are actively working on methods for controlling the robust solution radius sequence $\{\alpha_k\}$ for our next paper. One approach is by controlling the approximate PPM step size, with a smaller step size leading to finer $\alpha_k$ granularity. In practice, we can take initial large approximate PPM steps, and take finer approximate PPM steps once we enter a neighborhood of robust solutions with good efficiency-robustness trade-off for finer robust solution granularity.



**Action taken**: We give a new closed-formed solution to the robust solution radius $\alpha_k(\omega_k)$ for Theorem 1, in `appendix E`. We thank the reviewers for this comment as it opens up an important new research direction we hope to build upon the current paper.

---

### Meta-Review · Area_Chair_NYjX · 2024-12-23

**Metareview:**

A proximal point method based method is proposed to generate a series of Pareto efficient solutions with computational efficiency. However, the analysis is limited to a particular form of loss functions and is not obvious to the reviewers how to generalize to more general cases. Numerical results do not seem to be very convincing either.

**Additional Comments On Reviewer Discussion:**

The reviews are unanimously marginal reject. The authors provided comprehensive rebuttals, but they are not able to convince the reviewers.

---

### Decision · Program_Chairs · 2025-01-22

Reject